# CRTransSar: A Visual Transformer Based on Contextual Joint Representation Learning for SAR Ship Detection

Runfan Xia [1,2,3], Jie Chen [1,2,3,*], Zhixiang Huang [1,2], Huiyao Wan [1,2,3], Bocai Wu [3], Long Sun [3,4,5], Baidong Yao [3], Haibing Xiang [3] and Mengdao Xing [4,5]

1   Information Materials and Intelligent Sensing Laboratory of Anhui Province, Anhui University,
    Hefei 230601, China; p20301160@stu.ahu.edu.cn (R.X.); zxhuang@ahu.edu.cn (Z.H.);
    p19201033@stu.ahu.edu.cn (H.W.)
2   Key Laboratory of Intelligent Computing & Signal Processing, Ministry of Education, Anhui University,
    Hefei 230601, China
3   38th Research Institute of China Electronics Technology Group Corporation, Hefei 230601, China;
    18110995593@189.cn (B.W.); sl99goal@163.com (L.S.); yao1984@mail.ustc.edu.cn (B.Y.);
    xianghb@irsa.ac.cn (H.X.)
4   National Lab of Radar Signal Processing, Xidian University, Xi'an 710126, China; xmd@xidian.edu.cn
5   Collaborative Innovation Center of Information Sensing and Understanding, Xidian University,
    Xi'an 710126, China
*   Correspondence: jiechen@ustc.edu

**Abstract:** Synthetic-aperture radar (SAR) image target detection is widely used in military, civilian and other fields. However, existing detection methods have low accuracy due to the limitations presented by the strong scattering of SAR image targets, unclear edge contour information, multiple scales, strong sparseness, background interference, and other characteristics. In response, for SAR target detection tasks, this paper combines the global contextual information perception of transformers and the local feature representation capabilities of convolutional neural networks (CNNs) to innovatively propose a visual transformer framework based on contextual joint-representation learning, referred to as CRTransSar. First, this paper introduces the latest Swin Transformer as the basic architecture. Next, it introduces the CNN's local information capture and presents the design of a backbone, called CRbackbone, based on contextual joint representation learning, to extract richer contextual feature information while strengthening SAR target feature attributes. Furthermore, the design of a new cross-resolution attention-enhancement neck, called CAENeck, is presented to enhance the characterizability of multiscale SAR targets. The mAP of our method on the SSDD dataset attains 97.0% accuracy, reaching state-of-the-art levels. In addition, based on the HISEA-1 commercial SAR satellite, which has been launched into orbit and in whose development our research group participated, we released a larger-scale SAR multiclass target detection dataset, called SMCDD, which verifies the effectiveness of our method.

**Keywords:** transformer; deep learning; SAR target detection; multiscale learning; ship detection

## 1. Introduction

Synthetic-aperture radar (SAR) is an active microwave sensor that produces all-weather earth observations without being restricted by light and weather conditions. Compared with optical remote sensing images, SAR has significant application value. In recent years, SAR target detection and recognition have been widely used in military and civilian fields, such as military reconnaissance, situational awareness, agriculture, forestry management and urban planning. In particular, future war zones will extend from the traditional areas of land, sea and air to space. As a reconnaissance method with unique advantages, synthetic-aperture radar satellites may be used to seize the right to control information on future war zones and even play a decisive role in the outcome of these

wars. SAR image target detection and recognition is the key technology with which to realize these military and civilian applications. Its core idea is to efficiently filter out regions and targets of interest through detection algorithms, and accurately identify their category attributes.

By contrast, from optical images, the imaging mechanism of SAR images is very different. SAR targets have characteristics such as strong scattering, unclear edge contour information, multiscale, strong sparseness, weak, small, sidelobe interference, and complex background. The SAR target detection and recognition tasks present huge challenges. In recent years, many research teams have also conducted extensive research on the above-mentioned difficulties. For SAR target imaging problems, phase modulation from a moving target's higher-order movements severely degrades the focusing quality of SAR images, because the conventional SAR ground moving target imaging (GMTIm) algorithm assumes a constant target velocity in high-resolution GMTIm with single-channel SAR. To solve this problem, a novel SAR-GMTIm algorithm [1] in the compressive sensing (CS) framework is proposed to obtain high-resolution SAR images with highly focused responses and accurate relocation. To improve moving target detectors, one study proposed a new moving target indicator (MTI) scheme [2] by combining displaced-phase-center antenna (DPCA) and along-track interferometry (ATI) sequentially to reduce false alarms compared to MTI via either DPCA or ATI. As shown by the simulation results, the proposed method can not only reduce the false alarm rate significantly, but can also maintain a high detection rate. Another study proposed a synthetic-aperture radar (SAR) change-detection approach [3] based on a structural similarity index measure (SSIM) and multiple-window processing (MWP) The work proposed by focusing on SAR imaging [2] can be found in [1]. The main focus of these studies is on the detection of moving SAR targets [3] and changes in SAR images, while that of our study is SAR target detection.

The use of a detector with constant false-alarm rate (CFAR) [4] is common in radar target detection. Constant false-alarm rate detection is an important part of automatic radar target detection. It can be used as the first step in extracting targets from SAR images and it is the basis for further target identification. However, traditional methods rely too much on expert experience to design manual features, which have great feature limitations. The traditional methods are also difficult to adapt to SAR target detection in complex scenes and cannot be used for large-scale practical applications. Based on traditional feature-extraction target-detection methods, the histogram-of-oriented-gradient (HOG) feature is a feature descriptor used for object detection in computer vision and image processing. HOG calculates histograms based not on color values but on gradients. It constructs features by calculating and counting the histograms of gradient directions in local areas of the image. HOG features combined with support-vector-machine (SVM) classifiers have been widely used in SAR image recognition. In recent years, with the development of computer vision, convolutional neural networks have been applied to SAR image detection, and a large number of deep neural networks have been developed, including AlexNet [5], VGGNet [6], ResNet [7], and GoogLeNet [8]. Additionally, methods such as Faster R-CNN [9], SSD [10], and YOLO V3 [11] are also widely used in SAR image recognition. Moreover, we mainly rely on the advantages of CNN because it is highly skilled in extracting local feature information from images with more refined local attention capabilities. However, because of the large downsampling coefficient used in CNN to extract features, the network misses small targets. In addition, a large number of studies has shown that the actual receptive field in CNN is much smaller than the theoretical receptive field, which is not conducive to making full use of context information. CNN's feature capturability is unable to extract global representations. Although we can enhance CNN's global capturability by continuously stacking deeper convolutional layers, this results in a number of layers that are too deep, too many parameters for the model to learn, difficulty in effectively converging, and the possibility that the accuracy may not be greatly improved. Additionally, the model is too large, the amount of calculation increases sharply, and it becomes difficult to guarantee timeliness.

In recent years, the use of a classification and detection framework with a transformer [12] as the main body has received widespread attention. Since Google proposed bidirectional encoder representation from transformers (BERT) [13], the BERT model has also been developed, and the structure that plays an important role in BERT includes a transformer. Generalized autoregressive pretraining for language understanding (XLNET) [14] and other models have since emerged. BERT's core has not changed and still includes a transformer. The first vision transformer (ViT) for image classification was proposed in [15] and obtained the best results in optical natural scene recognition. Network models, such as detection transformer (DETR) [16] and Swin Transformer [17], with a transformer utilized for the main body, have appeared in succession.

Swin Transformer is currently mainly used in image classification, optical object detection, and the instance segmentation of natural scenes in the field of computer vision. In the field of remote sensing, the Swin Transformer is mainly used in image segmentation [18] and semantic segmentation [19]. We investigated the papers in this area in detail, and did not find any research work in the field of SAR target detection. We can transfer the entire framework to target segmentation and transfer work, which is also a focus of our future work, at a later date.

The successful application of a transformer in the field of image recognition is mainly due to three advantages. The first advantage includes the ability to break through the RNN model's limitation, enabling it to be calculated in parallel. The second advantage is that compared with CNN, the number of operations required to calculate the association between two positions does not increase with distance. The third advantage is that self-attention enables it to produce more interpretable models. We can check the attention distribution from the model. Each attention head can learn to perform different tasks. Compared with the CNN architecture, the transformer has better global feature capturabilities. Therefore, due to the key technical difficulties in the above-mentioned SAR target detection task, this paper combines the global context information perception of a transformer and the local information feature extractability of CNN that is oriented to the SAR target detection task, and innovatively proposes a context-based joint visual transformer framework for representation learning, referred to as CRTransSar. This is the first framework attempt in the field of SAR target detection. The experimental results from the SSDD and self-built SAR target dataset show that our method achieves higher precision. This paper focuses on the optimization design of the backbone and neck parts of the target detection framework. Therefore, we take the cascaded mask r-cnn framework as the basic framework of our method, and our method can be used as a functional module that is flexibly embedded in any other target detection frame. The main contributions of this paper include the following:

1.  First, to address the lack of global long-range modeling and perception capabilities of existing CNN-based SAR target detection methods, we designed an end-to-end SAR target detector with a visual Transformer as the backbone.
2.  Secondly, we incorporated strategies such as multi-dimensional hybrid convolution and self-attention, and constructed a new visual transformer backbone based on contextual joint representation learning, called CRbackbone, to improve the contextual salient feature description of multi-scale SAR targets.
3.  In addition, to better adapt to multi-scale changes and complex background disturbances, we constructed a new cross-resolution attention enhancement neck, called CAENeck, which can guide the multi-resolution learning of dynamic attention modules with little increase in computational complexity.
4.  Furthermore, we constructed a large-scale multi-class SAR target detection benchmark dataset. The source data were mainly from HISEA-1, China's first commercial remote sensing SAR satellite, developed by our research group.

## 2. Related Work

### 2.1. SAR Target Detection Algorithm

In traditional ship SAR target detection, the use of a detector with a constant false-alarm rate (CFAR) [4] is a common method for radar target detection. It can be used as the first step in extracting targets from SAR images and is the basis for further target identification. Chen et al. [20] proposed a histogram-based CFAR (H-CFAR) method, which directly uses the gray histogram of the SAR image and combines it with CFAR to successfully achieve ship target detection. Li et al. [4] proposed an improved super-pixel level CFAR detection method, which uses weighted information entropy (WIE) to describe the statistical characteristics of super-pixels and better distinguishes between targets and cluttered super-pixels. With the development of computer vision, convolutional neural networks have been applied to the detection of SAR images, and a large number of deep neural networks have emerged, such as AlexNet, VGGNet, ResNet, and GoogLeNet, which also enable Faster R-CNN, SSD, and YOLO V3. These neural networks are widely used in SAR image recognition. Roughly divided into two-stage detection methods, such as Mask R-CNN [21] and Faster R-CNN, and single-stage detection methods, such as YOLO V3 and SSD, transformers are added to combine with CNN, and deep-level extraction features are more suitable for SAR targets.

The two-stage detection method joins the fully connected segmentation subnet after the basic feature network and the original classification and regression task are divided into three tasks: classification, regression, and segmentation. These tasks are applied to SAR target detection to improve ship recognition accuracy. Its working principle is divided into four stages. First, a set of basic volumes, relu activation functions and pooling layers are used to extract features. Next, the feature maps are passed into the subsequent RPN and the fully connected layer. The RPN network is used to generate region proposals. Next, roi pooling of this layer collects the feature maps input by the convolutional layer and the proposals generated by the RPN network before passing them into the following fully connected layer to determine the target category. Finally, the proposal and features are used. The maps calculate the category of the proposal, and the bounding box regression is again used to obtain the detection frame's final precise position.

Single-stage detection methods, such as SSD [10], mainly detect specific targets directly from many dense anchor points and use features of different scales to predict the object. The main idea is to uniformly conduct dense sampling at different positions of the picture. Different sampling approaches can be used, including scale and aspect ratio. Next, CNN is used to extract features and directly perform classification and regression. The introduction of the YOLO series improves the detection speed.

Existing deep learning-based SAR ship detection algorithms have huge model sizes and very deep network scales. A series of algorithms are proposed by Xiaoling Zhang's team, ShipDeNet-20 [22] is a novel SAR ship detector, built from scratch; it is lighter than most algorithms and can be applied effectively to hardware transplantation. The detection accuracy of SAR ships is reduced due to the huge imbalance in the number of samples in different scenarios. Thus, to solve this problem, the authors of [23] proposed a balance scene learning mechanism (BSLM) for offshore and inshore ship detection in SAR images. In addition, the authors of [24] proposed a novel approach for high-speed ship detection in SAR images based on a grid convolutional neural network (G-CNN). This method improves the detection speed by meshing the input image, inspired by the basic principle of YOLO, and using depthwise separable convolution. However, existing most studies improve detection accuracy at the expense of detection speed. Thus, to solve this problem, HyperLi-Net was proposed [25] for high-accuracy and high-speed SAR ship detection. In addition, a novel high-speed SAR ship detection approach mainly using a depthwise separable convolution neural network (DS-CNN) was proposed [26]. In this approach, we integrated multi-scale detection mechanism, concatenation mechanism and anchor box mechanism to establish a brand-new light-weight network architecture for high-speed SAR ship detection. There are still some challenges hindering accuracy improvements for

SAR ship detection, such as complex background interferences, multi-scale ship feature differences, and indistinctive small ship features. Therefore, to address these problems, a novel quad feature pyramid network (Quad-FPN) [27] is proposed for SAR ship detection.

### 2.2. Transformer

Since the emergence of the attention mechanism and its high-quality performance in natural language processing, researchers have tried to introduce this attention mechanism into computer vision. Currently, however, research is mainly focused on optical natural image scenes in the field of image detection. Applying a transformer in the field of vision has recently become increasingly popular. Vision transformers [15] enable it to simultaneously learn low-level features and high-level semantic information by combining convolutional and regular transformers [12]. Experiments have proven that after replacing the last convolution module of Resnet [6] with a visual transformer, the number of parameters is reduced, and the accuracy is improved. DETR [16] uses a complete transformer to build an end-to-end target detection model. The largest highlight is the decoder. The original decoder is used to predict and generate sentence sequences, but in the target detection task, the input of the decoder is 0. The object vector outputs the object category and coordinates after FFN. The DETR model is simple and straightforward, except that the model abandons the manual method of designing anchors. Small objects have less pixel information and are easily lost in the downsampling process. For example, ships at sea are small in size. To address the detection problem of such obvious object size differences, the classic method is the image pyramid used for multiscale change enhancement, but this involves a considerable amount of calculation. However, multiscale detection is becoming increasingly important in target detection, especially for small targets.

In general, there are two main model architectures in the related work that use a transformer in computer vision (CV). One is a transformer-only structure [11], and the other is a hybrid structure that combines the backbone network of CNN with a transformer. In [15], a vision transformer was proposed for image classification for the first time. This research shows that dependence on CNN is not necessary. When directly applied to a sequence of image blocks, the transformer can also perform image classification tasks well. The research is based on a large amount of data for model pretraining and migration to multiple image recognition benchmark datasets. The results show that the vision transformer (ViT) model is comparable to the current optimal convolutional network. As a result, the computing resources required for its training are greatly reduced. The research specifically divides the image into multiple image patches and uses the linear embedding sequence of these image patches as the input for the transformer. Subsequently, the token processing method in the natural language processing (NLP) field is used to process the image block and train the image classification model in a supervised manner. When training with a medium-scale dataset (such as ImageNet), the model produces unsatisfactory results. This seemingly frustrating result is predictable. The transformer lacks some of the inherent inductive biases of CNN, such as translation, degeneration, and locality. Thus, after training with insufficient data, the transformer cannot generalize well. However, if the model is trained on a large dataset (14–300 m image), the situation is quite different. The study found that large-scale training outperforms inductive bias. When pretraining on a large enough data scale and migrating to tasks with fewer data points, the transformer can achieve excellent results.

### 2.3. Related Datasets in the Field of SAR Target Detection

On 1 December 2017, at the BIGSARDATA conference held in Beijing, China, a dataset SSDD [28] for ship target detection in SAR images was disclosed. SSDD is the first public dataset in this field. As of 25 August 2021, from 161 papers on deep learning-based SAR ship detection, 75 used SSDD as the training and test data, accounting for 46.6%, which shows the popularity and significance of SSDD in the SAR remote sensing community. The datasets used in other papers are the other five public datasets proposed in recent years, namely the SAR-Ship dataset released by Wang et al. in 2019, the AIR SARShip-1.0 released

by Sun et al. [29] HRSID released in 2020, and LS SSDD-v1.0, released by Zhang et al. [30] in 2020. The original paper of SSDD used a random ratio of 7:1:2 to divide the dataset into training set, validation set, and test set. However, this random partitioning mechanism leads to great uncertainty over the samples in the test set, resulting in different results when using the same detection algorithm for multiple training and testing. This is because the number of samples in SSDD is too small, only 1160, and random division may destroy the distribution consistency between the training and test sets. Similar to HRSID [31] and LS-SSDD-v1.0, here, images containing land are considered as near-shore samples, while other images are considered as far-sea samples. The numbers of near-shore and far-ocean samples were highly unbalanced (19.8% and 80.2%, respectively), a phenomenon consistent with the fact that the oceans cover much more of the Earth's surface than land. In the SSDD dataset, there are a total of 1160 images and 2456 ships, with an average of 2.12 ships per image, and the dataset will continue to expand in the future. Compared with the PASCAL VOC [32] dataset, which features 20 categories of objects, SSDD has fewer pictures, but the category is only ships, so it is enough to train the detection model.

The HRSID dataset was released by Su Hao from the University of Electronic Science and Technology of China in January 2020. HRSID is a dataset for ship detection, semantic segmentation, and instance segmentation tasks in high-resolution Sar images. The dataset contains a total of 5604 high-resolution SAR images and 16,951 ship instances. The ISSID dataset borrows from the construction process of the Microsoft common objects in context (COCO) [33] dataset, including SAR images at different resolutions, polarization, sea state, sea area, and coastal ports. This dataset is the benchmark against which the researchers evaluate their methods. For HRSID, the resolutions of the SAR images are: 0.5 m, 1 m, and 3 m, respectively.

## 3. The Proposed Method

This paper combines the respective advantages of the transformer [12] and CNN architectures, and is oriented to the SAR target detection task. Thus, we innovatively propose a visual transformer SAR target detection framework based on contextual joint-representation learning, called CRTransSar. The overall framework is shown in Figure 1. This is the first framework attempt in the field of SAR target detection.

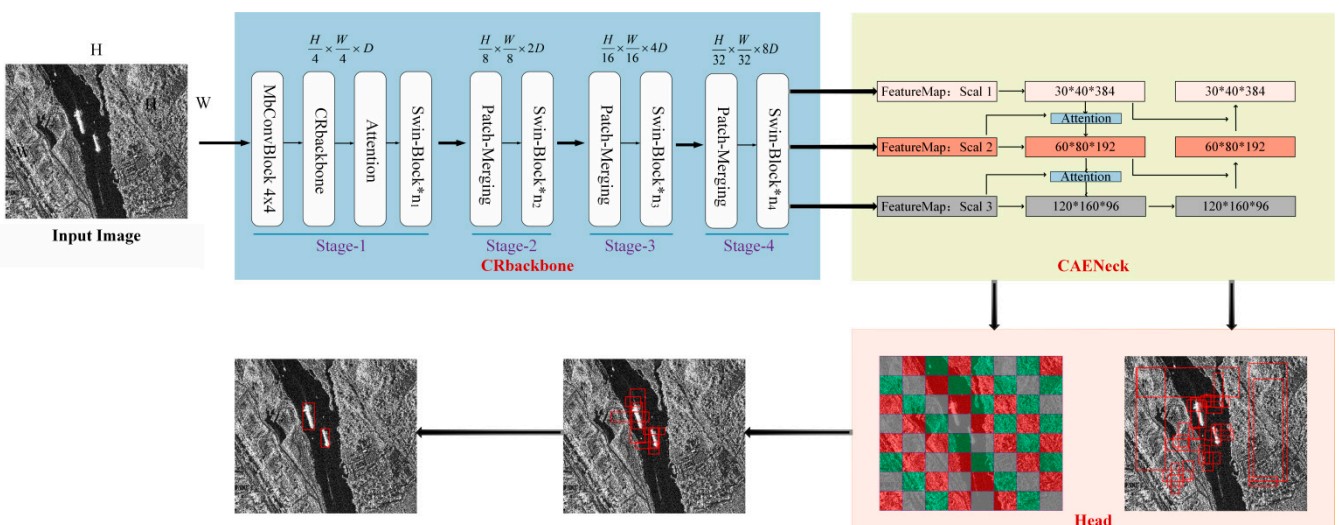

**Figure 1.** Overall architecture of the CRTransSar network.

### 3.1. The Overall Framework of Our CRTransSar

First, based on the cascade mask r-cnn two-stage model as the basic architecture, this paper innovatively introduces the latest Swin Transformer architecture as the backbone, introduces the local feature extraction module of CNN, and redesigns a target detection

framework. The design of the framework can fully extract and integrate the global and local joint representations.

Furthermore, this paper combines the respective advantages of a Swin Transformer and CNN to design a brand-new backbone, referred to as CRbackbone. Thus, the model can make full use of contextual information, perform joint-representation learning, extract richer contextual feature information, and improve the multi-characterization and description of multiscale SAR targets.

Finally, we designed a new cross-resolution attention enhancement Neck, CAENeck. A feature pyramid network [34] is used to convey strong semantic features from top to bottom, enhancing the two-way multiscale connection operation through top-down and bottom-up attention, while also aggregating the parameters from different backbone layers to different detection layers, which can guide the multi-resolution learning of dynamic attention modules with little increase in computational complexity.

As shown in Figure 1, CRTransSar is mainly composed of four parts: CRbackbone, CAENeck, RPN-Head, and Roi-Head. First, we used our designed CRbackbone to extract features from the input image and performed a multiscale fusion of the obtained feature maps. The bottom feature map is responsible for predicting small targets and the high-level feature map is responsible for predicting large targets. The RPN module receives the multiscale feature map and starts to generate anchor boxes, generating nine anchors corresponding to each point on the feature map, which can cover all possible objects on the original image. Using a $1 \times 1$ convolution to make prediction scores and prediction offsets for each anchor frame, all the anchor frames and labels were matched. Next, we calculated the value of IOU to determine whether the anchor frame belonged to the background or the foreground. Here, we establish a standard to distinguish the samples. The positive sample and the negative sample, after the above steps, obtain a set of suitable proposals. The received feature map and the above proposal are passed into ROI pooling for unified processing, and then finally passed to the fully connected RCNN network for classification and regression.

### 3.2. Backbone Based on Contextual Joint Representation Learning: CRbackbone

Aiming at the strong scattering, sparseness, multiscale, and other characteristics of SAR targets, this paper combines the respective advantages of transformer and CNN architectures to design a target detection backbone based on contextual joint representation learning, called CRbackbone. It performs joint representation learning, extracts richer contextual feature salient information, and improves the feature description of multiscale SAR targets.

First, we used the Swin Transformer, which currently performs best in NLP and optical classification tasks, as the basic backbone. Next, we incorporated CNN's multiscale local information acquisition and redesigned the architecture of a Swin Transformer. Influenced by the latest EfficientNet [35] and inspired by the architecture of CoTNet [36], we introduced multidimensional hybrid convolution in the patchembed part to expand the receptive field, depth, and resolution, which enhanced the feature perception domain. Furthermore, the self-attention module was introduced to strengthen the comparison between different windows on the feature map, and for contextual information exchange.

### 3.2.1. Swin Transformer Module

For SAR images, small target ships in large scenes easily lose information in the process of downsampling. Therefore, we use a Swin Transformer [17]. The framework is shown in Figure 2. The transformer has general modeling capabilities and it is complementary to convolution. It also has powerful modeling capabilities, better connections between vision and language, a large throughput, and large-scale parallel processing capabilities. When a picture is input into our network, first the transformer [11] is used to process the image because we need to use all of the means that can be processed to divide the picture into tokens similar to NLP with the high-resolution characteristics of the image. The language

difference leads to a layered transformer whose representation is calculated by moving the window. By limiting self-attention calculations to non-overlapping partial windows while allowing cross-window connections, the shifted window scheme leads to higher efficiency. This layered architecture has the flexibility of modeling at various scales and has linear computational complexity relative to the image size. This is an improvement to the vision transformer.

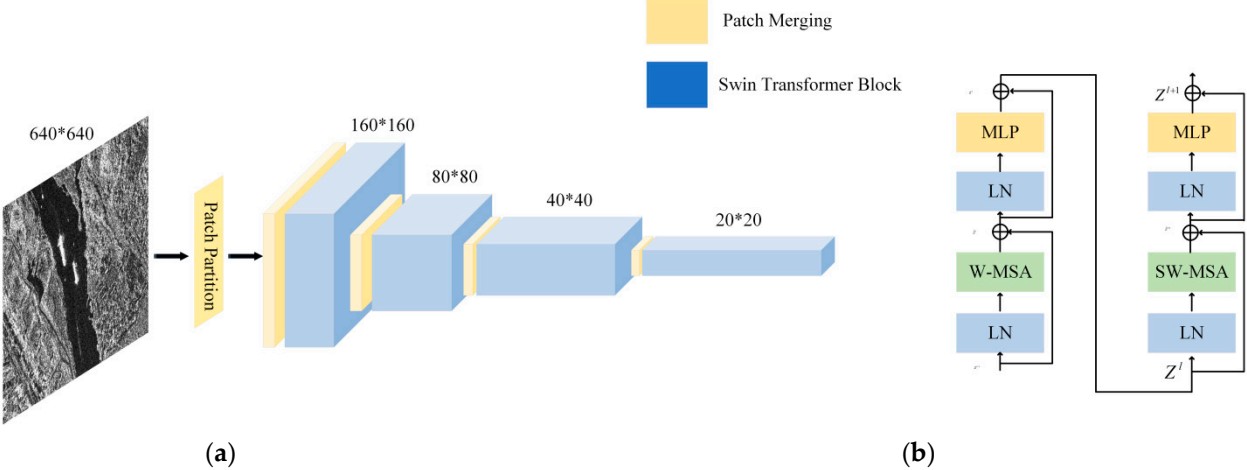

**Figure 2.** Overall architecture of Swin Transformer. (**a**) Swin Transformer structure diagram. (**b**) Swin Transformer blocks.

The vision transformer always focuses on the patch that is segmented at the beginning and does not perform any operations on the patch in the subsequent process. Thus, it does not affect the receptive field. A Swin Transformer is processed when a window is enlarged; subsequently, the calculation of self-attention is calculated in units of windows. This is equivalent to introducing locally aggregated information, which is very similar to the convolution process of CNN. The step size is the same as the size of the convolution kernel; thus, the windows do not overlap. The difference is that CNN performs the calculation of convolution in each window, and each window finally obtains a value, which represents the characteristics of this window. The Swin Transformer performs the self-attention calculation in each window and obtains an updated window. Next, through the patch merging operation, the window is merged, and the merged window continues to perform self-calculation. The Swin Transformer places the patches of the surrounding four windows together in the process of continuous downsampling, and the number of patches decreases. In the end, the entire image has only one window and seven patches. Therefore, we believe that downsampling means reducing the number of patches, but the size of the patches increases, which increases the receptive field.

As illustrated in Figure 3, the first module uses a regular window partitioning strategy, which starts from the top-left pixel, and the $8 \times 8$ feature map is evenly partitioned into $2 \times 2$ windows o $4 \times 4$ (M = 4) in size. Next, the next module adopts a windowing configuration that is shifted from that of the preceding layer by displacing the windows by (M/2, M/2) pixels from the regularly partitioned windows. With the shifted window-partitioning approach, consecutive Swin Transformer blocks are computed as:

$$\hat{\mathbf{z}}^l = \text{W-MSA}\left(\text{LN}\left(\mathbf{z}^{l-1}\right)\right) + \mathbf{z}^{l-1} \quad \mathbf{z}^l = \text{MLP}(\text{LN}(\hat{\mathbf{z}}^l)) + \hat{\mathbf{z}}^l \quad \hat{\mathbf{z}}^{l+1} = \text{SW-MSA}(\text{LN}(\mathbf{z}^l)) + \mathbf{z}^l \quad \mathbf{z}^{l+1} = \text{MLP}(\text{LN}(\hat{\mathbf{z}}^{l+1})) + \hat{\mathbf{z}}^{l+1} \quad (1)$$

where $\hat{\mathbf{z}}^l$ and $\mathbf{z}^l$ denote the output features of the SW-MSA module and the MLP module for block, respectively; W-MSA and SW-MSA denote window-based multi-head self-attention using regular and shifted window partitioning configurations, respectively.

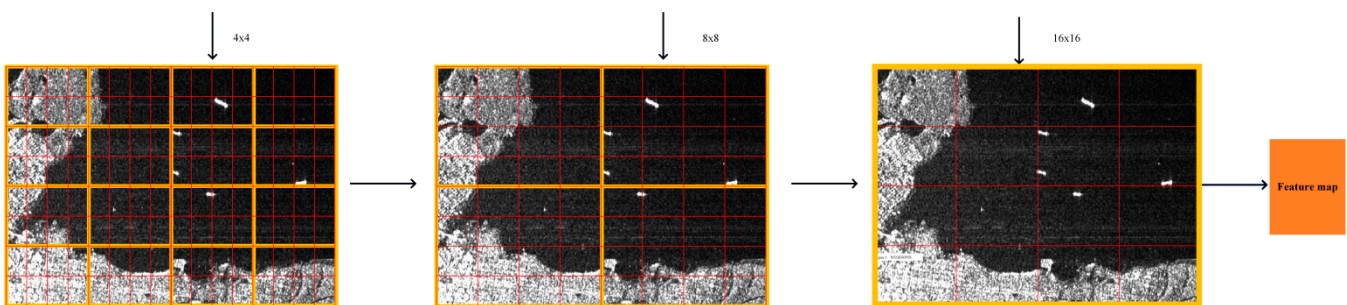

**Figure 3.** Swin Transformer sliding window. In the figure, 4, 8, 16 represent the number of patches.

A Swin Transformer performs self-attention in each window. Compared with the global attention calculation performed by a transformer, we assume that the complexity of a known MSA is the square of the image size. According to the complexity of an MSA, we can conclude that the complexity is $(3 \times 3)^2 = 81$. The Swin Transformer calculates self-attention in each local window (the red part). According to the complexity of MSA, we can see that the complexity of each red window is $1 \times 1$ squared, which is 1 to the fourth power. When there are nine windows, the complexity of these windows is summed, and the final complexity is nine, which is a greatly reduced figure. The calculation formulas for the complexity of an MSA and W-MSA are expressed by Formulas (2) and (3).

$$\Omega(MSA) = 4\text{hw}C^2 + 2(hw)^2C \tag{2}$$

$$\Omega(W\text{-}MSA) = 4\text{hw}C^2 + 2M^2hwC \tag{3}$$

Although computing self-attention inside the window may greatly reduce the complexity of the model, different windows cannot interact with each other, resulting in a lack of expressiveness. To better enhance the performance of the model, shifted-windows attention is introduced. Shifted windows alternately move between successive Swin Transformer blocks.

### 3.2.2. Self-Attention Module

Due to its spatial locality and other characteristics in computer vision tasks, CNN can only model local information and lacks the ability to model and perceive long distances. The use of a Swin Transformer introduces a shifted-window partition to improve this defect. The problem of information exchange between different windows is not limited to the exchange of local information. Furthermore, based on multihead attention, this paper takes into account the CotNet [36] contextual attention mechanism and proposes to integrate the attention module block into the Swin Transformer. The independent Q and K matrices are connected to each other. After the feature extraction network moves to patchembed, the feature map of the input network is 640*640*3. However, the length and width of the input data are not all 640*640. Next, we determine whether it is an integer multiple of 4 according to the length and width of the feature map to determine whether to pad the length and width of the feature map, followed by two convolutional layers. The feature channel changes from the previous 3 channels to 96 channels, and the feature dimension also changes to 1/4 of the previous dimension. Finally, the size of the attention module is 160*160*96, and the size of the convolution kernel is $3 \times 3$, as shown in Figure 4. The feature dimension and feature channel of the module remain unchanged, which strengthens the information exchange between the different windows on the feature map.

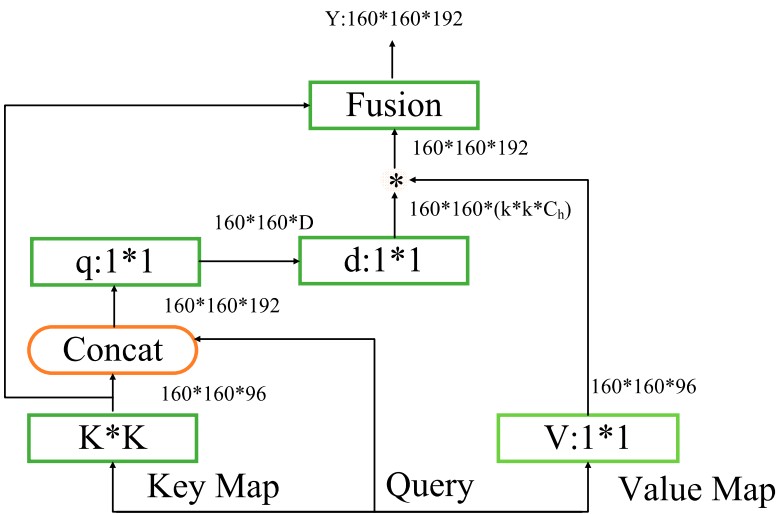

**Figure 4.** Self-attention module block.

The first step is to define three variables: Q = X, K = X, and V = XW$_v$. V is subjected to $1 \times 1$ convolution processing, then K is the grouped convolution operation of K × K and is recorded as the Q matrix and concat operation. The result after the concat performs two $1 \times 1$ convolutions and the calculation are shown in formula 4.

The self-attention module first encodes the contextual information of the input keys through $3 \times 3$ convolution to obtain a static contextual expression $K^1$ about the input; the encoded keys are further concatenated with the input query and the dynamic multi-head attention matrix is learned through two consecutive $1 \times 1$ convolutions. The resulting attention matrix is multiplied by the input values to obtain a dynamic contextual representation $K^2$ about the input. The fusion result of static context and dynamic context expression is used as output O. The architecture of the self-attention module is shown in Figure 5.

$$A = [K^1, Q]W_\theta W\delta \qquad (4)$$

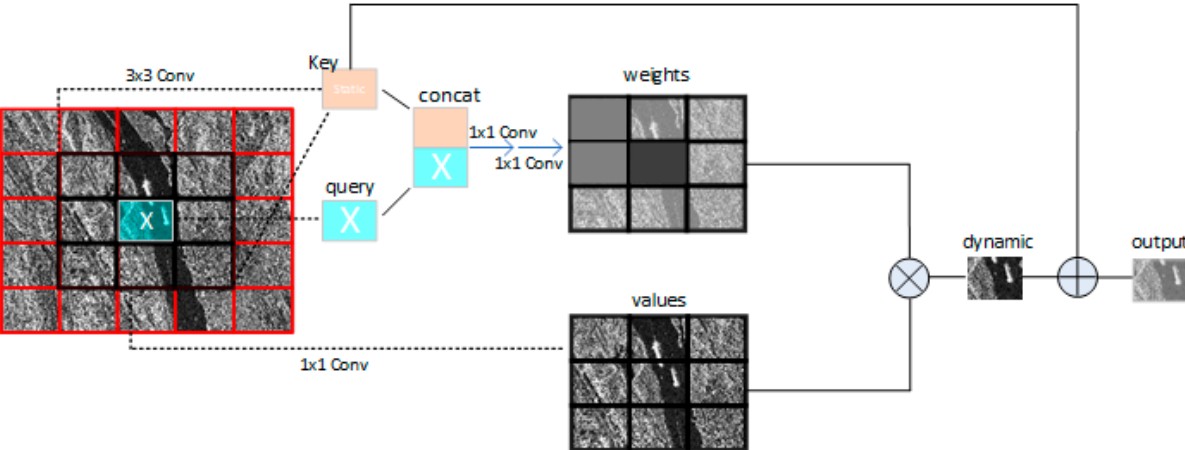

**Figure 5.** Architecture of self-attention module.

Here, *A* does not just model the relationship between *Q* and *K*. Thus, through the guidance of context modeling, the communication between each part is strengthened, and the self-attention mechanism is enhanced.

$$K^2 = V \otimes A \qquad (5)$$

$$O = K^2 \oplus K^1 \tag{6}$$

Unlike the traditional self-attention mechanism, the self-attention module block structure of this paper combines contextual information and self-attention. Unlike the latest global self-attention mechanism, HOG-ShipCLSNet [37] and PFGFE-Net [38] are distinguish the characteristics of different scales and different polarization modes, so as to ensure sufficient global responses to comprehensively describe SAR ships. The specific process is to first calculate $\Phi$ and $\theta$ through two $1 \times 1$ convolutional layers, and it is used to characterize feature A through a $1 \times 1$ convolutional layer. The $1 \times 1$ convolutional layer is used to characterize feature $g(\cdot)$. We then obtain the similarity f by matrix multiplication $\theta^T \Phi$, and, finally, f through a softmax function/layer with a sigmoid activation is multiplied by g to obtain the self-attention output. Furthermore, to make the output $y_i$ match the dimension of the input x to facilitate the follow-up element-wise adding operation, we add an extra $1 \times 1$ convolutional layer to achieve the dimension shaping. This is because in the embedded space, the number of convolution channels is c/2, which is not equal to the number of input channels c. This process is similar to the function of the residual or skip connections in ResNet, which can be described by $1 \times 1$. The weight matrix of the convolution layer is multiplied by $y_i$ and then added to the input.

### 3.2.3. Multidimensional Hybrid Convolution Module

To increase the receptive field according to the characteristics of the SAR target, this section describes the proposed method in detail. The feature extraction network proposed in this paper is based on a Swin Transformer in order to improve the backbone. The CNN convolution is integrated into the patchembed module with the attention mechanism, and it is reconstructed. The entire feature extraction network structure diagram is shown in Figure 6. Affected by the efficient network [35], a multidimensional hybrid convolution module is introduced in the patchembed module. The reason why we introduced this network is that according to the mechanism characteristics of CNN, the more convolutional layers are stacked, the larger the receptive field of the feature maps. We used this approach to expand the receptive field and the depth of the network, and to increase the resolution to improve the performance of the network.

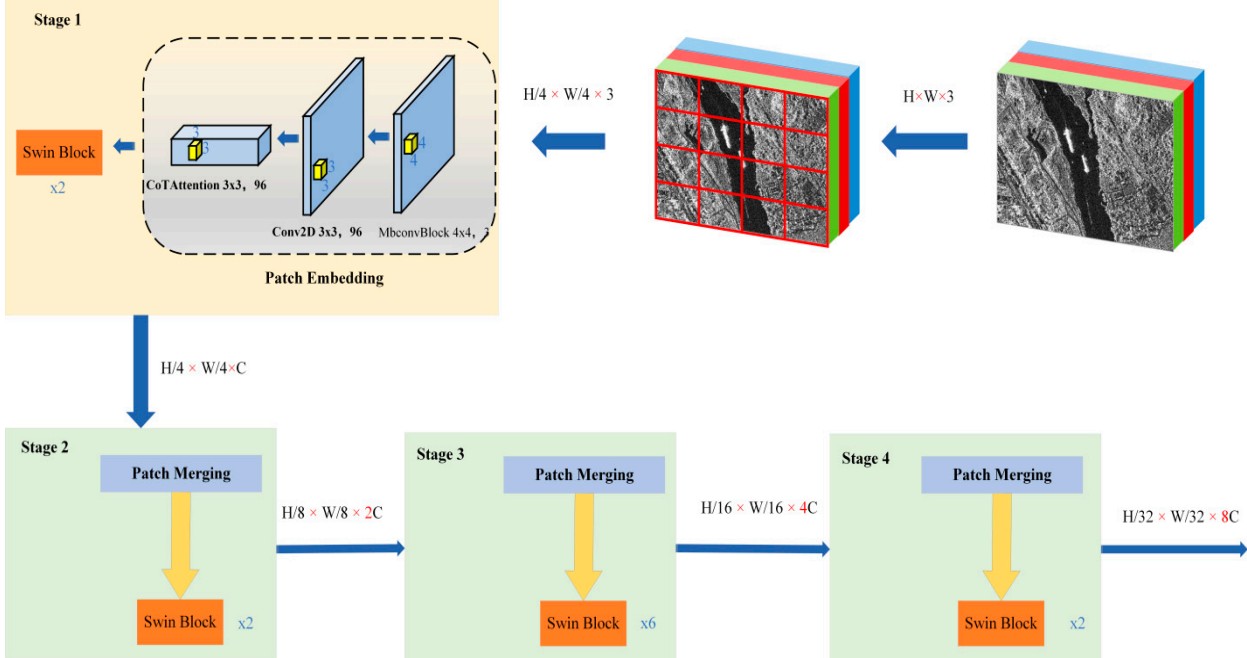

**Figure 6.** Overall architecture of CRbackbone.

When computing resources increase, if we thoroughly search for various combinations of the three variables of width, depth, and image resolution, the search space is infinite and the search efficiency is very low. The key to obtaining higher accuracy and efficiency is to balance the scaling ratios ($d$, $r$, $w$) of the three dimensions of network width, network depth, and image resolution using the combined zoom method:

$$\text{depth}: d = \alpha^\phi \text{width}: w = \beta^\phi \text{resolution}: r = \gamma^\phi \text{s. t.} \alpha \cdot \beta^2 \cdot \gamma^2 \approx 2\alpha \geq 1, \beta \geq 1, \gamma \geq 1 \tag{7}$$

$\alpha$, $\beta$, $\gamma$ are constants (not infinite because the three correspond to the amount of computation), which can be obtained by grid search. The mixing coefficient $\phi$ can be adjusted manually. If the network depth is doubled, the corresponding calculation amount will double, and the network width or image resolution will double the corresponding calculation amount, that is, the calculation amount of the convolution operation (FLOPS) is proportional to $d$, $\omega$ $\gamma^2$. There are two square terms in the condition. Under this constraint, after specifying the mixing coefficient $\phi$, the network calculation amount is about $2^\Phi$ times what it was before.

Now, we can integrate the above three methods and integrate the hybrid parameter expansion method. Although there is no lack of research in this direction about models, such as MobileNet [39], ShuffleNet [40], M-NasNet [41], etc., the model is compressed by reducing the amount of parameters and calculations. The model is also applied to mobile devices and edge devices, but the amount of parameters and calculations are considerably reduced at the same time. However, the accuracy of the model is greatly improved. The patchembed module mainly increases the channel dimensions of each patch, which are divided into non-overlapping patch sets by patch partitioning processing the input picture H × W × 3, which reduces the size of the feature map and sends it to the Swin Transformer block for processing. When each feature map is sent to patchembed's dimension 2 × 3 × H × W and then finally sent to the next module, the dimension is 2 × 3 × 96. When four downsamplings are achieved through the convolutional layer and the number of channels becomes 96, a layer of a multidimensional hybrid convolution module is stacked before the 3 × 3 convolution. The size of the convolution kernel is 4, keeping the number of channels fed into the convolution unchanged, which also increases the depth of the receptive field and the network. This improves the efficiency of the model.

### 3.3. Cross-Resolution Attention Enhancement Neck: CAENeck

This paper, inspired by the structure of SGE [42] and PAN [43], addresses the small targets in large scenes, including the strong scattering characteristics of SAR imaging and the characteristics of low discrimination between targets and backgrounds. This paper designs a new cross-resolution attention enhancement neck, called CAENeck.

The specific step is to divide the feature map into G groups according to the channel, and then to calculate the attention of each group. After global average pooling is performed on each group, g is obtained, and then g is a matrix multiplied with the original grouped feature map. Next, we proceed to the norm. Additionally, sigmoid was used to perform the operation, and the result obtained was the matrix multiplied by the original grouping feature map. The specific steps are shown in Figure 7.

The attention mechanism is added to connect the context information, and attention is incorporated at the top-to-bottom connection. This is to better integrate the shallow and deep feature map information and to better extract the features of small targets, along with the goals of the target. The positioning is shown in Figure 1. We upsample during the transfer process of the feature map from top to bottom. The size of the feature map increases. The deepest layer is strengthened by the attention module and concatenated with the feature map of the middle layer, which then passes through the attention module. A concat connection is formed with the most shallow feature map. The specific steps are as follows. The neck receives the feature maps of three scales; 30*40*384, 60*80*192, 120*160*96, and 30*40*384 are the deepest features, which are upsampled and pay attention to the force

enhancement operation, before being connected with 60*80*192. Finally, upsampling and attention enhancement are carried out to connect with the shallowest feature map.

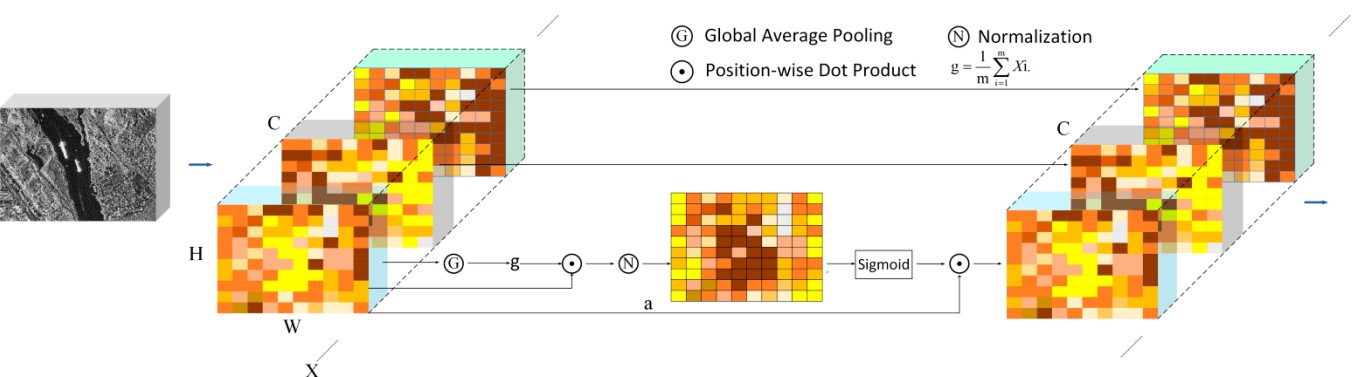

**Figure 7.** Neck attention enhancement module.

This series of operations is carried out from top to bottom. Next, bottom-up multiscale feature fusion is performed. Figure 1 shows the neck part. The SAR target is a very small target in a large scene, especially the marine ship target of the SSDD dataset. At sea, the ship itself has very little pixel information, and it easily loses the information of small objects in the process of downsampling. Although the high-level feature map is rich in semantic information for prediction, it is not conducive to the positioning of the target. The low-level feature map has little semantic information but is beneficial to the location of the target. The FPN [34] structure is a fusion of high-level and low-level from top to bottom. It is achieved through upsampling. The attention module is added during the upsampling process to integrate contextual information mining and self-attention mechanisms into a unified body. The ability to extract the information of the target location is enhanced. Furthermore, the bottom-up module has a pyramid structure from the bottom to the high level, which realizes the fusion of the bottom level and the high level after downsampling, while enhancing the extraction of semantic feature information. The small feature map is responsible for detecting large ships, and the large feature map is responsible for detecting small ships. Therefore, attention enhancement is very suitable for multiscale ship detection in SAR images.

### 3.4. Loss

The loss function is used to estimate the gap between the model output y and the true value y to guide the optimization of the model. This paper uses different loss functions in the head part. The specific formulas for the loss of the category in the RPN-head use cross-entropy loss, and the regression loss utilization function is as follows:

$$L(\{P_i\}, \{t_i\}) = \frac{1}{N_{class}}\sum_i L_{cls}(p_i, p_i^*) + \lambda \frac{1}{N_{reg}}\sum_i p_i^* L_{reg}(t_i, t_i^*) \tag{8}$$

where $\sum_i L_{cls}(p_i, p_i^*)$ represents the filtered anchor's classification loss, $P_i$ is the true value of each anchor's category, and $\sum_i p^* L_{reg}(t_i, t_i^*)$ is the predicted category of each anchor. Representing the loss of the regression, the function formula used for the regression loss is as follows:

$$L_{reg}(t_i, t_i^*) = \sum_{i \in x,y,w,h} smooth_{L1}(t_i - t_i^*) \tag{9}$$

$$smooth_{L1}(x) = \begin{cases} 0.5x^2 & if |x| < 1 \\ |x| - 0.5 & otherwise \end{cases} \tag{10}$$

## 4. Experiments and Results

This section evaluates our proposed detection method through experimental results. First, we use the SSDD dataset and the SMCDD dataset as experimental data. The SMCDD dataset provides some of the parameter settings of the experiment. The next part describes the influence of the attention enhancement backbone, the reconstruction of the patchembed module, and the multiscale attention enhancement neck. Finally, we compare our method with other methods to verify the effectiveness of our method. Our hardware platform is a personal computer with an Intel i5 CPU based on the mmdet [44] framework, an NVIDIA RTX2060 GPU, 8 GB of video memory, and an Ubuntu 18.04 operating system.

*4.1. Dataset*

4.1.1. SSDD Dataset

We used a remote sensing SAR image dataset. The SAR ship detection established in 2017 used a SSDD ship dataset, which sets the baseline of the SAR ship detection algorithm and is used by many other scholars. The SSDD dataset contains data in a variety of scenarios, including different polarization modes and scenarios. We used the same labeling method as the most popular PASCAL VOC dataset. A total of 1160 images and 2456 ships were included, with an average of 2.12 ships per image. Although the number of images was small, we used it as a benchmark for ship target detection performance. The ratio of the training image, verification image, and test image was 7:2:1. The SSDD dataset is shown in Table 1.

**Table 1.** SSDD dataset description.

| Category | Indicator |
|---|---|
| Scenes | RadarSat-2, TerraSAR-X, Sentinel-1 |
| Polarization | HH, VV, HV, VH |
| Resolution | 1–15 m |
| Number of pictures | 1160 |
| Number of ships | 2456 |

4.1.2. SMCDD Dataset

Our research group will soon release the SAR dataset, which contains data from the satellite HISEA-1 called SMCDD, as shown in Figure 8.

The HISEA-1 satellite is China's first commercial SAR synthetic-aperture radar satellite, jointly developed by the 38th Research Institute of China Electronics Technology Group Corporation, China Changsha Tianyi Space Science and Technology Research Institute Co., Ltd. (Changsha, China), as well as other units. Since its entry into orbit, the HISEA-1 has performed more than 1880 imaging tasks, obtaining 2026 striped images, 757 spotlight images, and 284 scanned images. The HISEA-1 has the ability to provide stable data services. The slice data of the SMCDD dataset we constructed are all from the SAR large scene image captured by HISEA-1.

Our SMCDD dataset contains four types of data: ship data, airplane data, bridge data, and oil tank data, as shown in Figure 9. The images we used were all large. There were four polarization modes, and, as a result, we cut them into 256, 512, 1024, and 2048 sizes. We used slices of 1024 and 2048 sizes and finally passed our screening and cleaning, leaving 1851 bridges, 39,858 ships, 12,319 oil tanks, and 6368 aircraft, as shown in the figure. Although the current version of the dataset is unbalanced, we will continue to expand the dataset in the future. We also verified the effectiveness of the method proposed in this paper through our dataset. The data information is shown in Table 2.

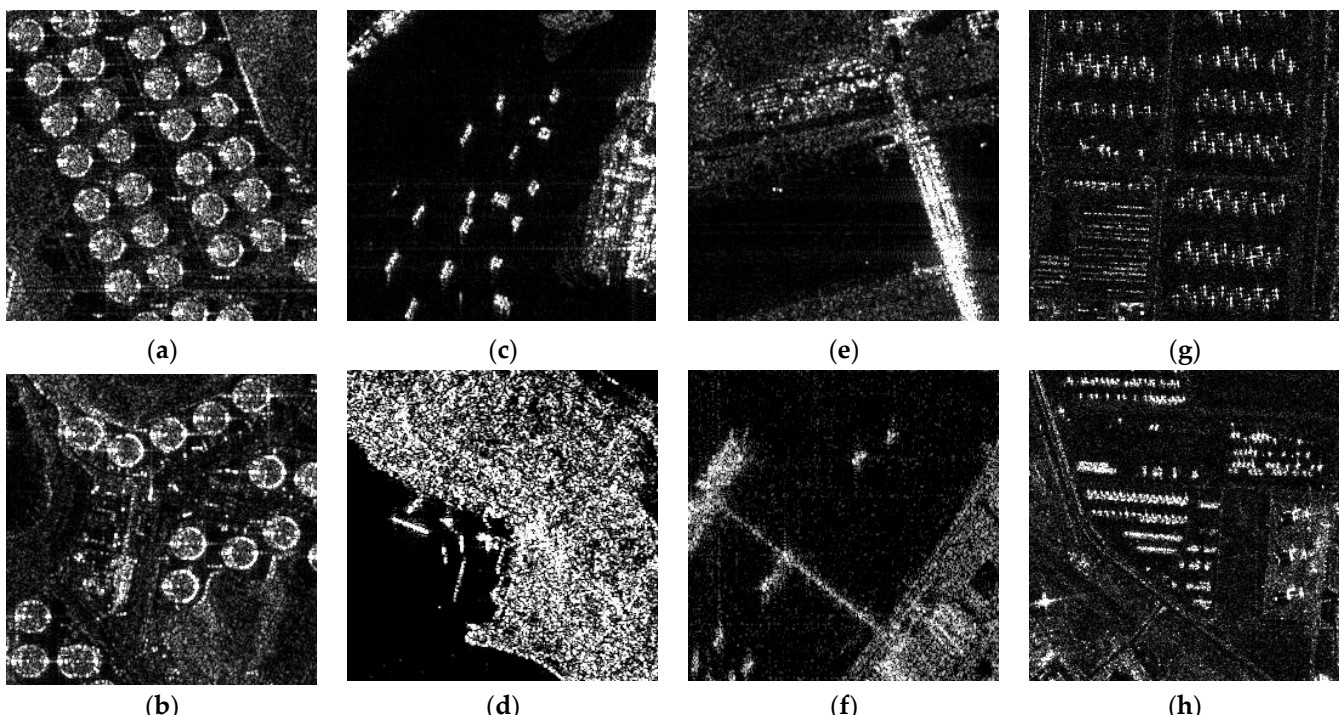

**Figure 8.** Some examples of the dataset SMCDD to be released by the research group. (**a**,**b**) are oil tanks; (**c**,**d**) are ship; (**e**,**f**) are bridges; (**g**,**h**) are aircraft.

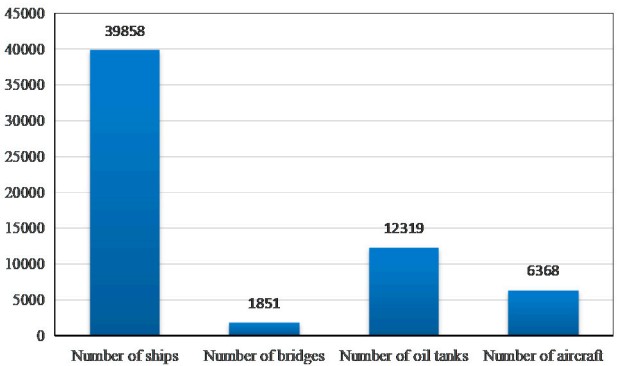

**Figure 9.** Description of the research group dataset SMCDD.

**Table 2.** Description of the research group dataset SMCDD.

| Category | Indicator |
|:---:|:---:|
| Scenes | HISEA-1 |
| Polarization | HH, VV, HV, VH |

In contrast to the existing open-source SAR target detection dataset, our SMCDD dataset has the following advantages:

(1) The existing SAR target detection dataset is only for ship detection, and our dataset categories are richer, covering aircraft, ships, bridges, and oil tanks.

(2) The existing SAR target detection data collection is small, which makes it difficult to support the effective training of large-scale models. Our data collection is larger, which can support the training and verification of large-scale models.

(3) Since our dataset covers different types of SAR target data, our dataset can be used as a verification library for research directions, such as multiclass detection and

recognition, long-tailed distribution (class imbalance), small sample detection and recognition, etc. This will greatly promote the overall development of the SAR target detection field.

(4) We cut our large-scene images into various sizes, such as 2048*2048, 1024*1024, 512*512, and 256*256. We filtered and cleaned the data, leaving 1851 bridges, 39,858 ships, 12,319 oil tanks, and 6368 aircraft. Although the current version of the dataset is unbalanced, we will continue to expand the dataset in the future. For larger targets, such as bridges, we need to choose more large slice samples, generally 1024*1024 or 2048*2048, so that the network can better train the data.

(5) We have large-scene SAR images of HISEA-1, which can provide a large amount of training data to improve the SAR target detection performance of the network in large scenes.

### 4.2. Setup and Implementation Details

This study used Python 3.7.10, PyTorch 1.6.0, CUDA 10.1, CUDNN 7.6.3, and MMCV1.3.1, and the results of our network pretraining model were Swin-T on ImageNet. A total of 500 epochs was set up for training in the entire network. Due to the limitations of computer hardware and the size of the network itself, the batch size was set to 2. Each training sent two images to the network for processing, and the AdamW optimizer was selected as the model. The initial learning rate was set to 0.0001, the weight attenuation was 0.0001, and strategies such as LoadImageFromFile, LoadAnnotations, RandomFlip, and AutoAugment were used to optimize the training pipeline, as well as to enhance the online data, which enhanced the robustness of the algorithm. We adjusted the image size and finally selected the most suitable size for the network proposed in this paper to be 640*640.

### 4.3. Evaluation Metric

To quantitatively evaluate the performance of the proposed cascade mask rcnn with the improved Swin Transformer as the backbone and CAENeck as the neck detection algorithm, the accuracy, recall rate, average accuracy (mAP) and F-measure (F1) were used as evaluation indicators. Accuracy refers to the rate of correct detection of ships in all detection results, and recall refers to the rate of correct detection of ships in all ground facts. The definition of precision and recall is as follows:

$$P = \frac{TP}{TP + FP} \tag{11}$$

$$R = \frac{TP}{TP + FN} \tag{12}$$

In the formula, *TP*, *FP*, and *FN* represent the positive samples predicted by the model as positive, the negative samples predicted by the model as positive, and the positive samples predicted by the model as negative, respectively. In addition, if the IoU between the predicted bounding box and the real bounding box is higher than the threshold of 0.5, the bounding box is recognized as a correctly detected ship. The precision recall (*PR*) curve shows the precision recall rate under different confidence thresholds. *MAP* is a comprehensive metric that calculates the average precision under the recall range [0, 1]. The definition of m*AP* is as follows:

$$\mathrm{m}AP = \int_0^1 P(R)\mathrm{d}R \tag{13}$$

In the formula, *R* is the recall value, which represents the precision corresponding to the recall. *F*1 evaluates the comprehensive performance of the detection network proposed in this paper by considering the accuracy and recall rate. *F*1 is defined as:

$$F1 = \frac{2 \times P \times R}{P + R} \tag{14}$$

*4.4. Analysis of Experimental Results*

4.4.1. Ablation Experiments

A. The Influence of CRbackbone on the Experimental Evaluation Index

During the experiment, we first added the improved network backbone to the network, and the neck part was consistent with the baseline. We compared the benchmark Swin Transformer as the backbone, and PAN as the neck, and evaluated the test indicators. The comparison results are shown in Table 3. We observed that adding the optimized backbone to the percentage of mAP (0.5) led to an improvement of 0.4%; the improvement of mAP (0.75) was 6.5%, and the recall rate was also improved, by 1.3%. Therefore, the improved backbone had a propelling effect on the optimization of the network.

**Table 3.** The influence of CRbackbone on the experimental evaluation index.

| Method | P | R | F1 | mAP$_{50}$ | mAP$_{75}$ |
|--------|-----|-----|-----|-------|-------|
| CRbackbone | 0.920 | 0.982 | 0.950 | 0.965 | 0.766 |
| Baseline | 0.912 | 0.969 | 0.940 | 0.961 | 0.701 |

B. The Influence of the CAENeck Module on the Experimental Evaluation Index

During the experiment, first added the improved network neck into the network, and the backbone part was consistent with the baseline. The lightweight attention-enhancement neck module was discarded to study its influence on the experiment, and an evaluation of the experimental indicators was carried out. The comparison results are shown in Table 4. We observed a 0.1% improvement in the percentage of mAP (0.5), a 2.5% improvement in mAP (0.75), and a 0.2% improvement in the recall rate. In the neck part, the detection performance also improved.

**Table 4.** The influence of CAENeck module on experimental evaluation index.

| Method | P | R | F1 | mAP$_{50}$ | mAP$_{75}$ |
|--------|-----|-----|-----|-------|-------|
| CAENeck | 0.918 | 0.971 | 0.944 | 0.962 | 0.726 |
| Baseline | 0.912 | 0.969 | 0.940 | 0.961 | 0.701 |

4.4.2. Experimental Comparison with Current Methods

To compare traditional methods and advanced methods, we adopted the same parameter settings to test and verify them. We propose to use the improved Swin Transformer as the backbone's cascade mask RCNN target detection network to verify and compare the SSDD dataset and the dataset to be released by our research group. The experimental results are shown in the following table.

The target detection model proposed in this paper achieved a substantial improvement over the SSDD dataset. The accuracy of mAP (0.5) reached 97.0%, the accuracy of mA (0.75) reached 76.2%, and the F1 was 95.3. It can be seen that through the improvement of the Swin Transformer, the integration of the CotNet attention mechanism, and the lightweight EfficientNet module in patchembed promoted the optimization of the backbone. The cross-resolution attention enhancement neck strengthened the characteristics of the different scales. The fusion of the maps and these several methods are of great help for detecting ships. We compared the two-stage, single-stage, and anchor-free methods. The experiments showed that the detection accuracy of the method proposed in this paper is generally higher than that of the two-stage methods, such as Faster RCNN (88.5%) and Cascade R-CNN [45] (89.3%). We also compared our results with those of single-stage yolov3 (95.1%), SSD (84.9%), and RetinaNet (90.5%) [46]. The experimental results were also higher than those of the single-stage detection algorithm, which shows that the transformer uses the attention mechanism. Powerful functions, cascaded local information, and enhanced multiscale fusion is more conducive to the detection of inshore vessels without the perception of noise or the identification of ships of different sizes. We also compared the most advanced

FCOS [47] and CenterNet [48] without anchor frame detection and Cascade R-CNN [45] and Libra R-CNN [49] with anchor frame detection, as shown in Tables 5 and 6. This paper also draws the PR curve to compare the difference between the different networks, in Figure 10.

**Table 5.** Comparison with the latest anchor-free target detection method.

| Method | P | R | F1 | mAP$_{50}$ | mAP$_{75}$ |
|---|---|---|---|---|---|
| FCOS [47] | 0.872 | 0.925 | 0.898 | 0.951 | – |
| CenterNet [48] | 0.803 | 0.933 | 0.863 | 0.945 | – |
| CRTransSar (Ours) | 0.925 | 0.983 | 0.953 | 0.970 | 0.762 |

**Table 6.** Comparison with the latest anchor-based target detection method.

| Method | P | R | F1 | mAP$_{50}$ | mAP$_{75}$ |
|---|---|---|---|---|---|
| Faster R-CNN [9] | 0.810 | 0.942 | 0.871 | 0.885 | 0.488 |
| Cascade Mask R-CNN | 0.840 | 0.940 | 0.887 | 0.902 | – |
| Cascade R-CNN [45] | 0.819 | 0.930 | 0.871 | 0.893 | – |
| YOLOV3 [11] | 0.873 | 0.960 | 0.914 | 0.951 | 0.659 |
| SSD [10] | 0.770 | 0.93 | 0.883 | 0.849 | – |
| RetinaNet [26] | 0.870 | 0.945 | 0.906 | 0.905 | 0.526 |
| Libra R-CNN [49] | 0.808 | 0.924 | 0.862 | 0.887 | – |
| CRTransSar (Ours) | 0.925 | 0.983 | 0.953 | 0.970 | 0.762 |

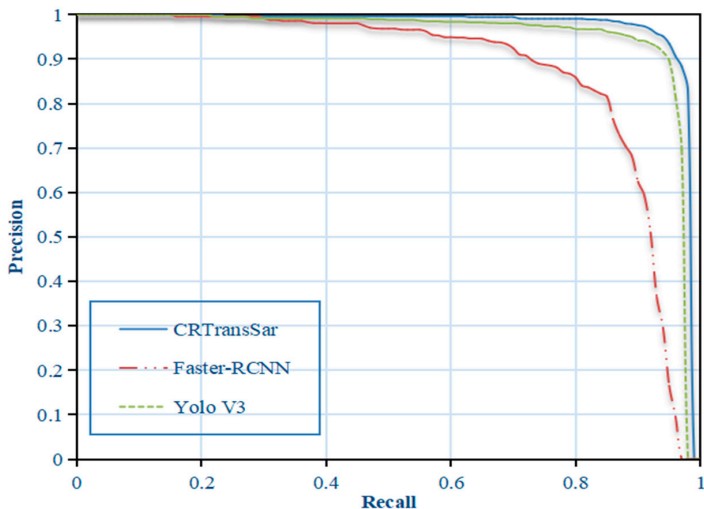

**Figure 10.** Comparison with the PR curve of the classic method.

The basic principle of CenterNet is that each target object is modeled as a center point to represent it. No candidate frame is required, nor is postprocessing, such as non-maximum suppression. CenterNet uses a fully convolutional network to generate a high-resolution feature map, classifies and judges each pixel of the feature map, and determines whether it is the center point of the target category or the background. This feature map gives each target the position of the center point of the object, the processing confidence in the center point of the target is 1, and the confidence of the background point is 0. Now, since there is no anchor box, there is no need to calculate the IoU between the anchor box and the bounding box to obtain positive samples to directly train the regressor. Instead, each point (located within the bounding box and having the correct class label) that is determined to be a positive sample is part of the regression of the bounding box size parameter.

This paper quotes the latest SAR target detection methods, FBR-Net [50], Center-Net++ [51], NNAM [52], DCMSNM [53], and DAPN [54]. Since the relevant papers do not have specific data divisions, this paper has no way to fully reproduce the results from other relevant papers. Therefore, this paper can only quote them. The results are compared horizontally, as shown in Table 7.

**Table 7.** Comparison with the latest SAR target detection method.

| Method | P | R | F1 | mAP |
|---|---|---|---|---|
| FBR-Net [50] | 0.928 | 0.940 | 0.934 | 0.941 |
| CenterNet++ [51] | 0.833 | 0.952 | 0.889 | 0.951 |
| NNAM [52] | 0.843 | 0.851 | 0.849 | 0.798 |
| DCMSNM [53] | 0.836 | 0.834 | 0.835 | 0.896 |
| DAPN [54] | 0.711 | 0.909 | 0.798 | 0.898 |
| CRTransSar (Ours) | 0.925 | 0.983 | 0.953 | 0.970 |

To demonstrate the robustness of our proposed algorithm, we conducted comparative experiments with low SNR on salt and pepper noise, random noise, and Gaussian noise. In the salt and pepper noise experiments, our method led to mAP of 94.8. The map of our method was 5.5% higher than Yolo v3 and 9.7% higher than Faster R-CNN. In the random noise experiments, our method led to mAP of 96.7. The map of our method was 3% higher than Yolo v3 and 11.1% higher than Faster R-CNN. In the Gaussian noise experiments, our method led to mAP of 95.8. The map of our method was 1% higher than Yolo v3 and 10.7% higher than Faster R-CNN. The experimental results, presented in Table 8, show that we produced a reliable performance for SAR target detection tasks in low SNR.

**Table 8.** Comparison with low SNR of other advanced methods.

| Method | Salt and Pepper Noise | | | | Gaussian Noise | | | | Random Noise | | | |
|---|---|---|---|---|---|---|---|---|---|---|---|---|
| | P | R | mAP | F1 | P | R | mAP | F1 | P | R | mAP | F1 |
| Faster R-CNN | 80.0 | 92.7 | 85.1 | 85.9 | 79.2 | 92.4 | 85.1 | 85.3 | 79.8 | 92.5 | 85.6 | 85.7 |
| YOLO | 84.3 | 93.4 | 89.3 | 88.6 | 89.4 | 97.2 | 94.8 | 93.1 | 88.7 | 97.2 | 93.7 | 92.7 |
| CRTransSar (Ours) | 90.2 | 96.7 | 94.8 | 93.3 | 90.0 | 97.8 | 95.8 | 93.7 | 91.3 | 98.2 | 96.7 | 94.6 |

In order for our proposed method to effectively solve the SAR target detection task, we also made corresponding experimental comparisons for the computational cost of the Swin Transformer. The FPS and parameter statistics of several representative target detection algorithms are shown in Table 9. Compared with the single-stage target detection algorithm, our parameter was 34M higher than YOLO V3, and the FPS was 28.5M lower, but the mAP was 1.9% higher than YOLO v3. Compared with two-stage target detection, the parameter amount was 52M higher than Faster R-CNN, and the FPS was 11.5M lower. Compared with Cascade R-CNN, the parameter amount was 8M higher and the FPS was 4.5M lower. Compared with Cascade Mask R-CNN, the number of parameters was 19M higher and the FPS was 7.5M lower. However, our mAP was 8.5% higher than that of Faster R-CNN, 7.7% higher than that of Cascade R-CNN, and 6.8% higher than that of Cascade Mask R-CNN. Because the overall architecture of the Swin Transformer is still relatively large, the large volume of Transformer is a general problem in this field, and we plan to make further improvements in model lightweighting and compression in the future.

**Table 9.** Compared with computational cost of other advanced methods.

| Method | mAP | R | FPS (img/s) | Parameter (M) |
|---|---|---|---|---|
| yolo v3 | 0.951 | 0.960 | 36 | 62 |
| Faster R-CNN [9] | 0.885 | 0.942 | 3 | 44 |
| RetinaNet | 0.905 | 0.945 | 3 | 77 |
| Cascade R-CNN | 0.893 | 0.930 | 12 | 88 |
| Cascade Mask R-CNN | 0.902 | 0.940 | 14 | 77 |
| CRTransSar (Ours) | 0.970 | 0.983 | 7.5 | 96 |

4.4.3. Comparison between Experimental Results of the SMCDD Data Set

We used state-of-the-art object detection methods to evaluate our self-built SMCDD dataset. We chose CRTransSar, RetinaNet, and YOLOV3 as our benchmark algorithms, as shown in Table 10. There was a large number of dense targets in the oil tanks and aircraft in the SMCDD data set, which posed great challenges to the detection. It can be seen from the data that CRTransSar's mAP reached 16.3, which was better than RetinaNet and yolov3, and it was also higher than these two models in Recall.

**Table 10.** Comparison results on the SMCDD data set.

| Method | mAP | R |
|---|---|---|
| RetinaNet [46] | 0.161 | 0.203 |
| YOLOV3 [11] | 0.150 | 0.128 |
| CRTransSar(Ours) | 0.163 | 0.250 |

*4.5. Visualization Result Verification and Analysis*

To verify the effectiveness of the method in this paper, we visualized the SSDD dataset and the dataset to be released by our own research group, and obtained satisfactory results. We randomly selected some near-shore and far-shore ships for inspection. It can be seen from the figure that the use of multiscale fusion feature maps can more effectively improve the results of SAR images in different scenes, meaning that the method proposed in this paper can extract features with rich semantic information, even from complex backgrounds near shore. This method can also eliminate the interference and accurately identify the place where the naked eye has difficulty distinguishing between the noise and the ship. It can also eliminate some marine object noise, such as ships in the distant sea, and can be accurately distinguished. We also accurately verified the ships photographed by HISEA-1.

(1) This section visually verifies the performance of the network from two datasets, which are divided into inshore and offshore sets. Figure 11 shows the visual verification of SSDD inshore ships. When there is a relatively small amount of dense ships, the network's detection performance is better, and it is not disturbed by shore noise. Figure 12 is the dataset to be released by our research group, which contains inshore ships photographed by the HISEA-1 satellite and high-resolution satellites.

(2) Figures 13 and 14 are the SSDD dataset of the far sea and the results of the identification of the offshore ships of the dataset to be released by our research group, respectively. Offshore, because the surrounding environment receives less noise interference, the recognition accuracy is higher than it is inshore. Therefore, almost all target ships can be accurately identified in the offshore scene.

(3) To demonstrate the object detection performance of our proposed method for large scenes, we selected our self-built SMCDD dataset as the inference dataset. Our original data were obtained from the 38th Research Institute of China Electronics Technology Group Corporation. Because the data belonged to a secret military institution, we signed a confidentiality agreement with them. The original image of the large scene was obtained by HISEA-1. However, to further demonstrate the effectiveness of our method, we used the sliced data of some large scenes with a size of 2048*2048. As shown in Figure 15, from the

visualization results, it can be seen that Figure 15a,f are missing detections in three places; Figure 15b,d feature one false detection.

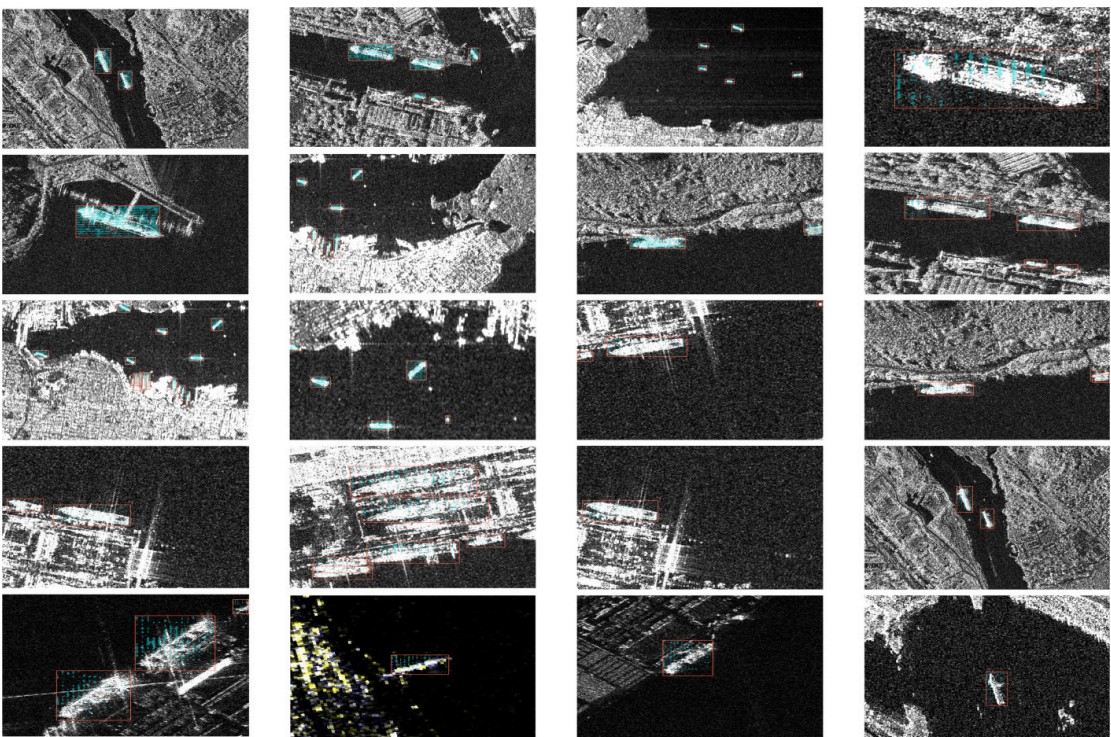

**Figure 11.** SSDD inshore inspection results. The red rectangular box is the correct visualization result of the CRTransSar method inshore on the SSDD dataset.

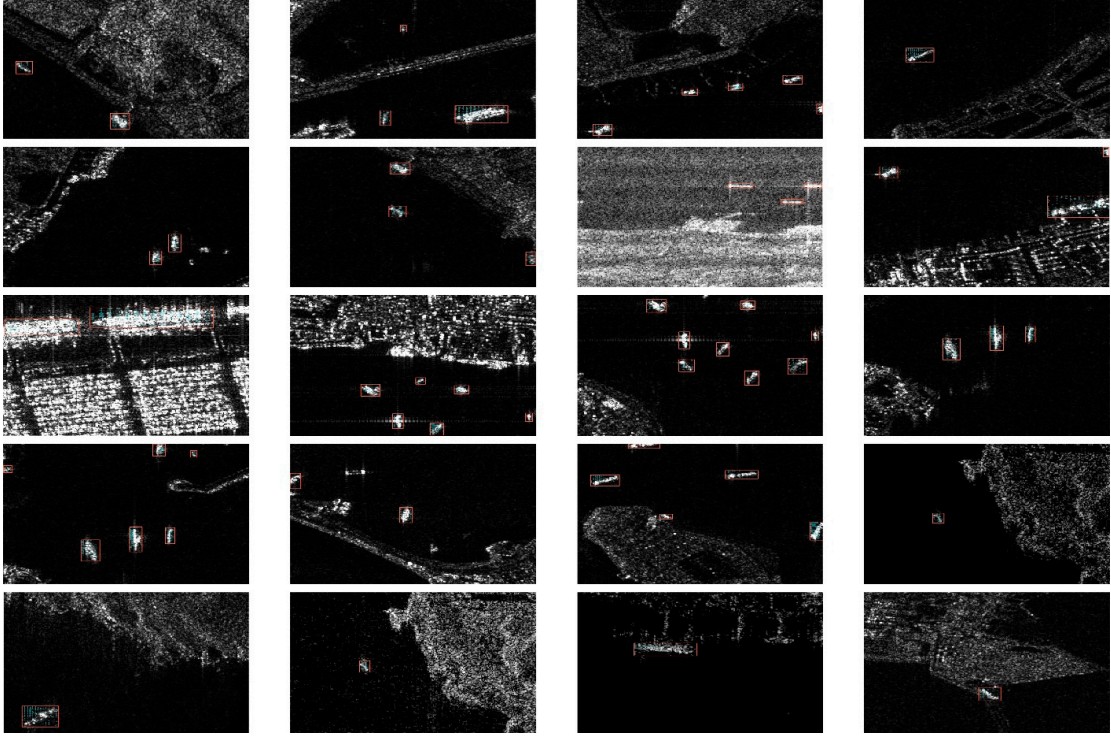

**Figure 12.** The results of verification of inshore taken by HISEA-1 Satellite. The red rectangular box is the correct visualization result of the CRTransSar method in the inshore scene of a port captured by HISEA-1 Satellite.

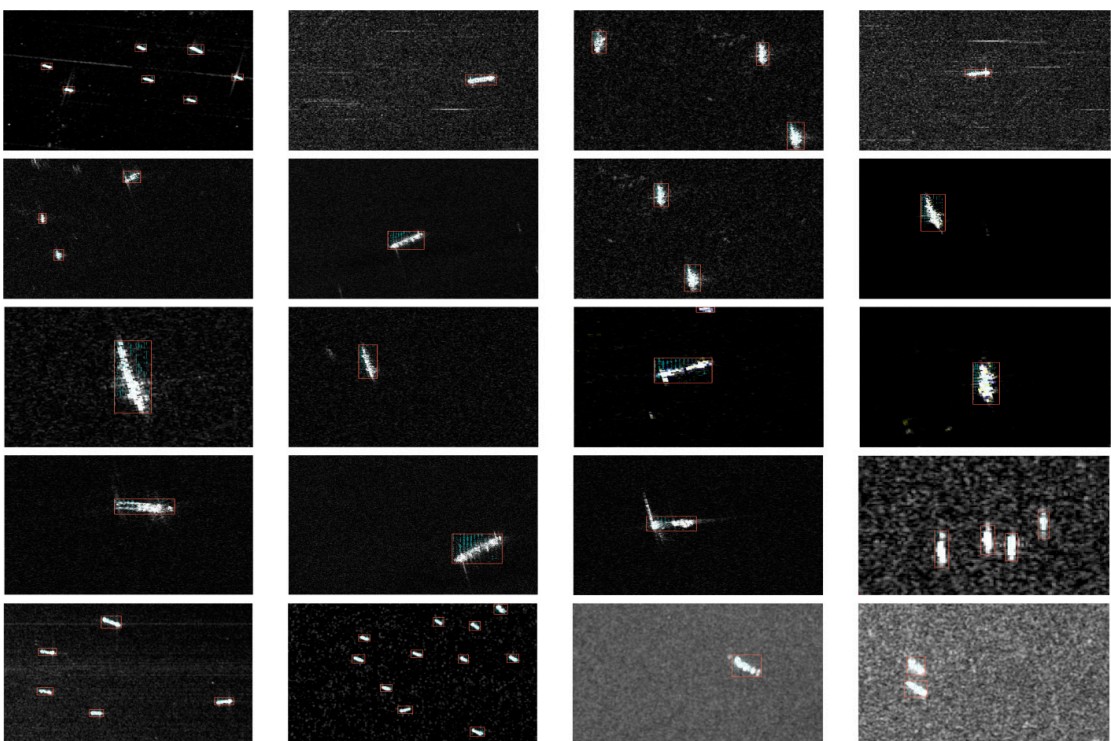

**Figure 13.** SSDD offshore ship identification result. The red rectangular box is the correct visualization result of the CRTransSar method offshore on the SSDD dataset.

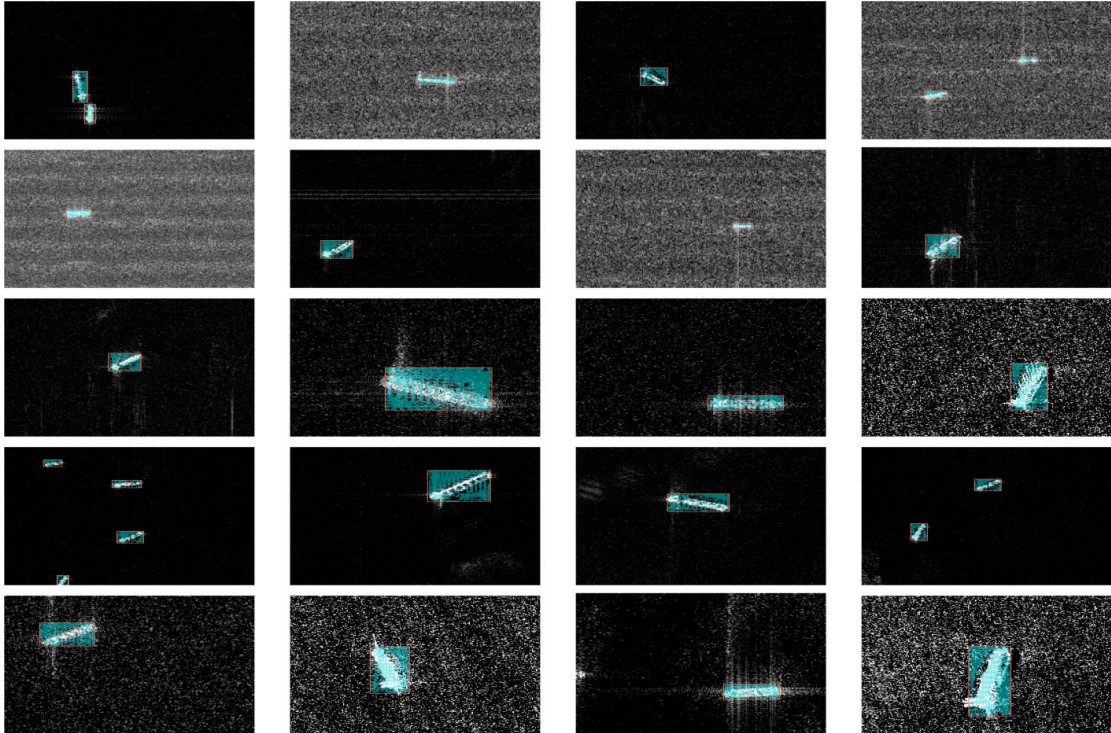

**Figure 14.** The results of verification of offshore ships taken by HISEA-1 satellites. The red rectangular box is the correct visualization result of the CRTransSar method in the offshore scene of a port captured by HISEA-1 Satellite.

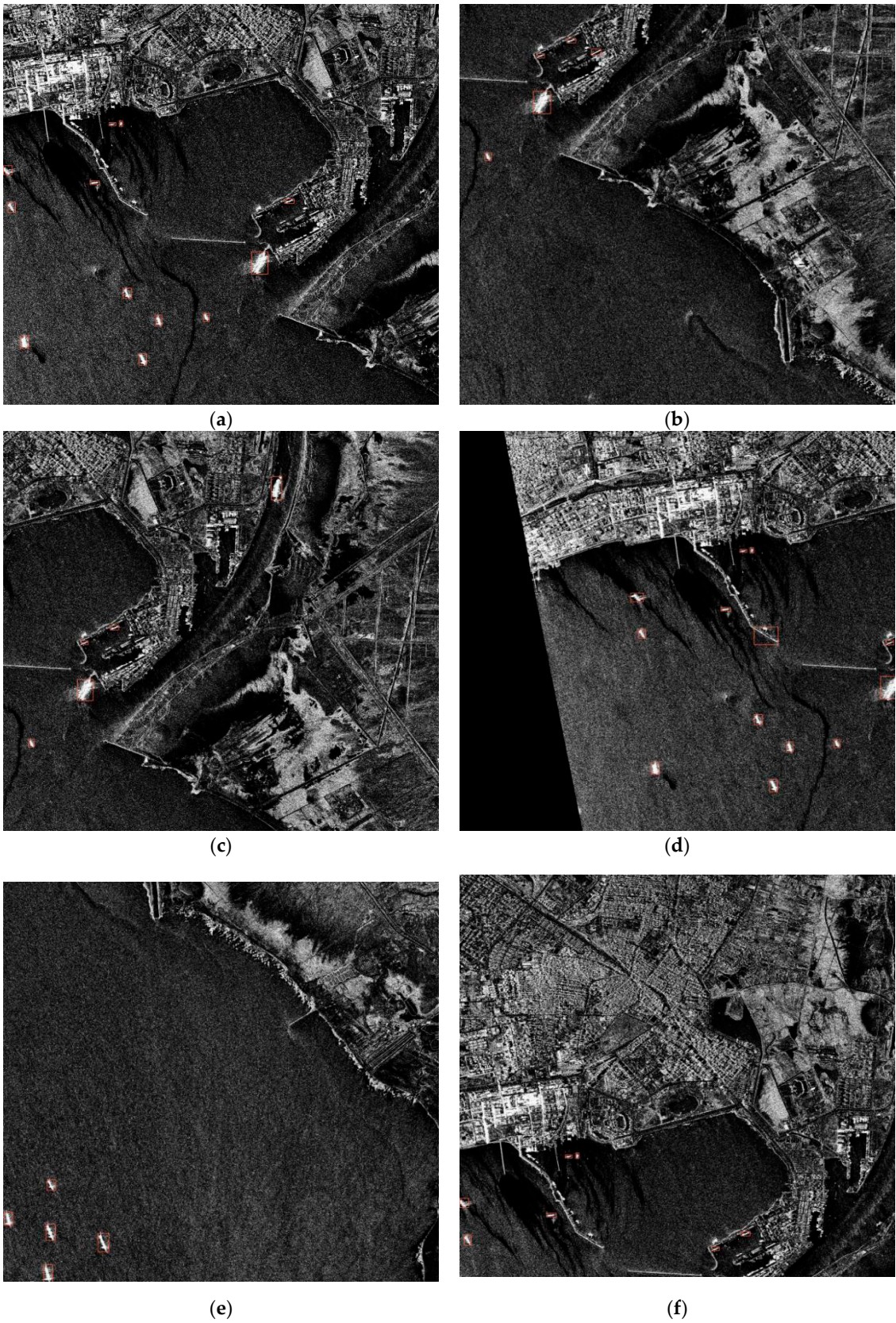

**Figure 15.** Visualization results of large scenes. (**a**–**f**) are slices of a large scene taken by HISEA-1 Satellite in a port. The red mark in the figure is the visualization result of the ships detected by CRTransSar in this scene.

## 5. Discussion

The four graphs in Figure 16 show some errors in the visualized results. It can be seen that (a) there are obvious detection frames to identify a ship, picture (b) has obvious undetected ships detected, and picture (c) has obvious detection frames to identify multiple ships, while one ship is detected by multiple ships. In (d), there are multiple ships that have not been recognized. We can solve the problem of the difficult identification of neighboring ships by segmentation, and introduce nonlocal mean models to highlight edge information.

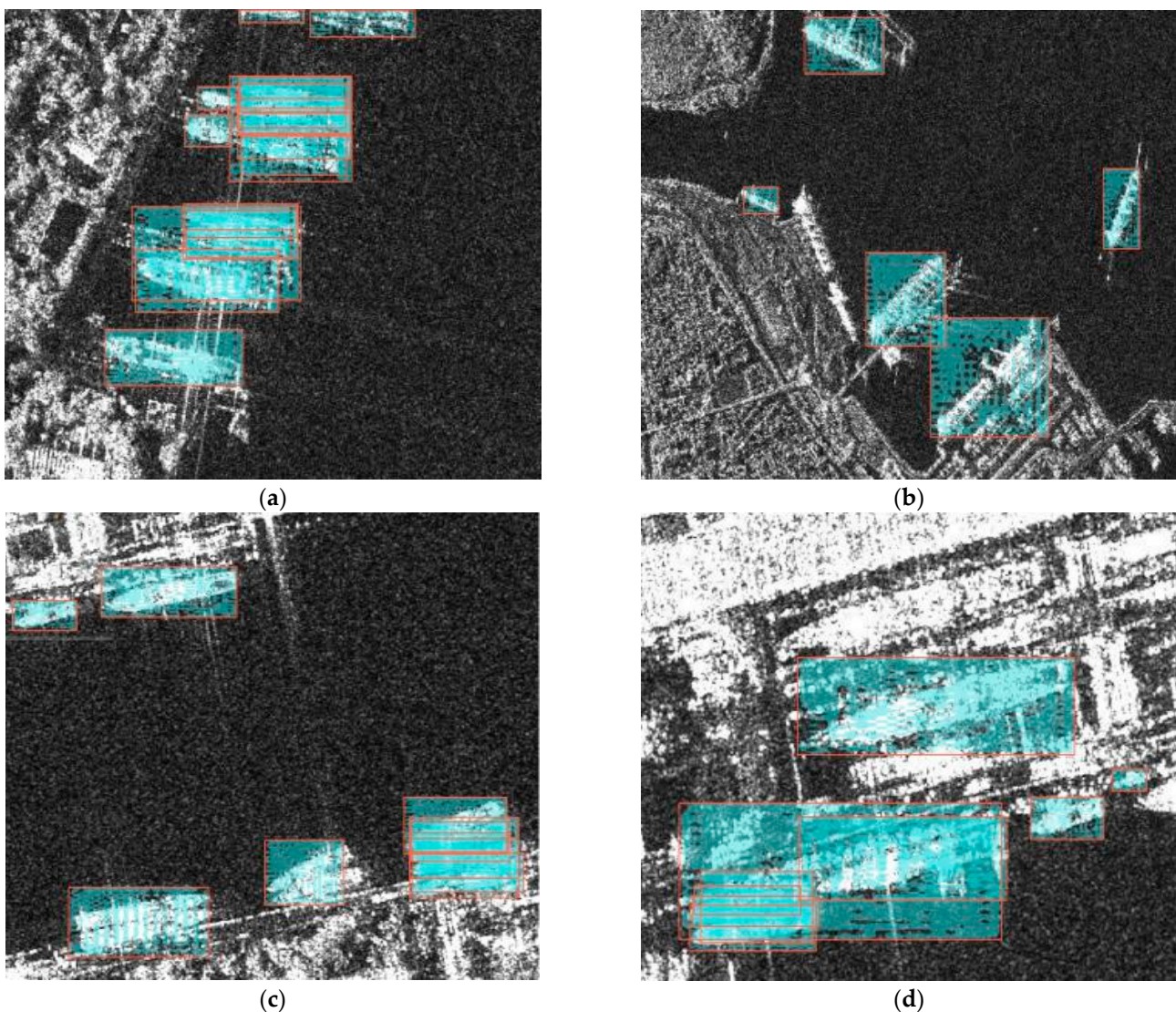

**Figure 16.** Visualization results of misdetected samples. (**a**–**d**) are the inshore visualization results of the CRTransSar method under the SSDD dataset. (**a**,**c**,**d**) are the visualization results of false-alarms. (**b**,**d**) are the visualization results of missed-detection.

## 6. Conclusions

SAR target detection has important application value in military and civilian fields. Aiming to overcome the difficulties of SAR targets, such as strong scattering, sparseness, multiscale, unclear contour information, and complex interference, we propose a visual transformer SAR target detection framework based on contextual joint representation learning, called CRTransSar. In this paper, CNN and the transformer are innovatively combined to improve the feature representation and the detectability of SAR targets in a balanced manner. This study was based on the use of a Swin Transformer and integrates CNN architecture ideas. We also redesigned a new backbone, named CRbackbone, which

makes full use of contextual information, conducts joint-representation learning, and extracts richer context-feature salient information. Furthermore, we constructed a new cross-resolution attention enhancement neck, called CAENeck, which is used to enhance the ability to characterize SAR targets at different scales.

We conducted related experiments on the SSDD dataset and SMCDD dataset, as well as verification experiments on the SSDD dataset and the SMCDD dataset to be released by our research group. We performed visual verification of the classification of near-shore vessels and high-water vessels in the verification experiment. The high-quality results prove the robustness and practicability of our method. In the comparison experiment on the two-stage and no-anchor frames, higher precision was achieved. The method proposed in this paper achieves 97.0% mAP (0.5) and 76.2% mAP (0.75). In future work, we will first standardize the SMCDD dataset of our research group and release it for download and use. In addition, we will introduce segmentation to detect densely adjacent ships and explore more efficient distillation methods that do not require time-consuming training. Combined with pruning methods, model compression will be more diversified and easier to transplant, as will the lightweight development of the network.

**Author Contributions:** Conceptualization, R.X.; methodology, R.X.; software, R.X.; validation, R.X.; formal analysis, R.X.; investigation, R.X.; resources, R.X.; data curation, R.X.; writing—original draft preparation, R.X.; writing—review and editing, R.X., J.C., Z.H. and H.W.; visualization, R.X.; supervision, J.C., Z.H., B.W., L.S., B.Y., H.X. and M.X.; project administration, J.C.; funding acquisition, J.C. All authors have read and agreed to the published version of the manuscript.

**Funding:** This work was supported in part by the National Natural Science Foundation of China under Grant 62001003, in part by the Natural Science Foundation of Anhui Province under Grant 2008085QF284, and in part by the China Postdoctoral Science Foundation under Grant 2020M671851.

**Institutional Review Board Statement:** Not applicable.

**Informed Consent Statement:** Not applicable.

**Data Availability Statement:** The data used in this study are open data sets. The dataset can be downloaded at https://pan.baidu.com/s/1paex4cEYdTMjAf5R2Cy9ng (2 February 2022).

**Acknowledgments:** We would like to thank the anonymous reviewers for their constructive and valuable suggestions on the earlier drafts of this manuscript.

**Conflicts of Interest:** The authors declare no conflict of interest.

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
