# Peer review of "CRTransSar: A Visual Transformer Based on Contextual Joint Representation Learning for SAR Ship Detection"

_remotesensing, doi:10.3390/rs14061488_

Round 1

Reviewer 1 Report

The authors proposed a framework of combined CNN and the transformer for the improvement of feature representation and the detectability of SAR targets. I appreciate the effort made by authors in writing the paper, however, the paper needs to a lot of modifications.

First, the paper is a bit long and the quality of presentation is low as its quantity, length, and number of words grew.  Here I mention some of the problems:

The authors should define all the acronyms just after the first use : CFAR, HOG, SVM, CV, NLP and so on

Line 105) It is highly recommended to express the contributions of the paper succinctly, using  bullet points and only one or two sentences for each item.

Line 149) Use the reference just after et al.. Also the same comment for the other parts of the paper. Also use reference for  PASCAL VOC at line 451.

Lines 186-187) "but currently, they are mainly focused on studying optical natural image scenes in the field of SAR remote sensing image detection" It is optical or SAR?

Lines 188-189) " Visual transformers[14] combine convolution and regular transformers[11] in order for convolution to learn low- level features and for transformers to learn high-level semantic information. " This is not clear.

Line 229) Consider changing the title. The proposed method or framework, conveys better than "Methods"

Lines 273-276)  It is repetitive. This is only one example of repetition in the manuscript. Try to omit the repetitive sentences from all parts of the paper.

Line 292) "Therefore, we introduce a Swin Transformer[16]" We use or we introduce?! The same comment for line 438 (we introduce the SSDD dataset)

English language: the paper needs to be read again and again. There are so many sentences in the paper that are vague or grammatically incorrect

Line 345) based on wheth er or not 4 can be divisible. 

Line 296) When a picture is input into our network, we want to send it to a transforms former[11]. We want to send is not suitable here.

Line 232) This is the first framework attempt in the... either framework or attempt seems to be unnecessary 

Line 412) SARS?

Lines 437-438) the sentence implies that SSDD will also be released by your research group in near future.

Line 448) sniper?

Typos and punctuations: The authors should have made a double check before submission, 

Line 31: Synthetic is missed

Line 357 is bold.

Lines 432  - 433) formulas are displaced

Lines 628-629) First,this & backbone,CRbackbone

Author Response

 Response to Reviewer 1 Comments

Dear professor,

Thank you for reading the paper and for your careful comments!

Furthermore, We have completed relevant revisions and improvements to our paper based on your valuable comments.

Best regards,

Corresponding Author: Jie Chen1, 2, 3

1Information Materials and Intelligent Sensing Laboratory of Anhui Province, Anhui University Hefei, 230601, China

2 Key Laboratory of Intelligent Computing & Signal Processing, Ministry of Education, Anhui University, Hefei, 230601, China

338th Research Institute of China Electronics Technology Group Corporation in Hefei, China

Point 1: The authors should define all the acronyms just after the first use: CFAR, HOG, SVM, CV, NLP and so on.

Response 1: Thanks for your valuable comments! We rechecked our manuscript and defined acronyms before they were first used. In addition, we also carefully checked the standardized expressions of other professional vocabulary in the manuscript.

Point 2: Line 105) It is highly recommended to express the contributions of the paper succinctly, using bullet points and only one or two sentences for each item.

Response 2: Thanks for your valuable comments! We have revised and streamlined based on your suggestions, and have reorganized the main contributions of our manuscript. The modified contents are as follows:

The main contributions of this paper include the following:

(1) First, to address the lack of global long-range modeling and perception capabilities of existing CNN-based SAR target detection methods, we design an end-to-end SAR target detector with a visual Transformer as the backbone.

(2) Secondly, we design strategies such as multi-dimensional hybrid convolution and self-attention, and construct a new visual transformer backbone based on contextual joint representation learning, called CRbackbone, to improve the contextual salient feature description of multi-scale SAR targets.

(3) In addition, to better adapt to multi-scale changes and complex background disturbances, we construct a new cross-resolution attention enhancement Neck, called CAENeck, which can guide the multi-resolution learning of dynamic attention modules with little increase in computational complexity.

(4) Furthermore, we construct a large-scale multi-class SAR target detection benchmark dataset. The source data are mainly from HISEA-1, China's first commercial remote sensing SAR satellite developed by our research group.

Point 3: Line 149) Use the reference just after et al. Also the same comment for the other parts of the paper. Also use reference for PASCAL VOC at line 451.

Response 3: Thank you for your valuable advice! We have carefully revised each of the above items based on your valuable suggestions. In addition, We also rechecked all literature citations in the manuscript to improve the accuracy of the manuscript.

Point 4: Lines 186-187) "but currently, they are mainly focused on studying optical natural image scenes in the field of SAR remote sensing image detection" It is optical or SAR?

Response 4: Thanks for your valuable comments! We have made changes to the corresponding parts of the manuscript and highlighted them.

Point 5: Lines 188-189)"Visual transformer[14] combine convolution and regular transformers[11] in order for convolution to learn low-level features and for transformers to learn high-level semantic information. " This is not clear.

Response 5: Thank you for your valuable comments! We have made changes to the corresponding sentences and highlighted them in the manuscript.

Point 6: Line 229) Consider changing the title. The proposed method or framework, conveys better than "Methods".

Response 6: Thank you for your valuable advice! We have made changes to the corresponding titles and highlighted them in the manuscript.

Point 7: Lines 273-276) It is repetitive. This is only one example of repetition in the manuscript. Try to omit the repetitive sentences from all parts of the paper.

Response 7: Thank you for your valuable comments! We have removed repeated examples in the manuscript and double-checked language expressions in the manuscript.

Point 8: Line 292) "Therefore, we introduce a Swin Transformer[16]" We use or we introduce?! The same comment for line 438 (we introduce the SSDD dataset).

Response 8: Thank you for your valuable comments! We have revised the corresponding words and highlighted them in the manuscript.

Point 9: English language: the paper needs to be read again and again. There are so many sentences in the paper that are vague or grammatically incorrect.

Response 9: Thank you for your valuable comments! We have made improvements to the sentences in the manuscript.

Point 10: Line 345) based on whether or not 4 can be divisible. 

Response 10: Thank you for your valuable advice! We have made changes to the corresponding sentences and highlighted them in the manuscript.

Point 11: Line 296) When a picture is input into our network, we want to send it to a transforms former[11]. We want to send is not suitable here.

Response 11: Thank you for your valuable comments! We have revised the corresponding words and highlighted them in the manuscript.

Point 12: Line 232) This is the first framework attempt in the... either framework or attempt seems to be unnecessary. 

Response 12: Thank you for your valuable comments! We have revised the corresponding words and highlighted them in the manuscript.

Point 13: Line 412) SARS?

Response 13: Thank you for your valuable advice! We have revised the corresponding words and highlighted them in the manuscript.

Point 14: Lines 437-438) the sentence implies that SSDD will also be released by your research group in near future.

Response 14: Thank you for your valuable advice! We have made changes to the corresponding sentences and highlighted them in the manuscript.

Point 15: Line 448) sniper?

Response 15: Thank you for your valuable advice! We have revised the corresponding words and highlighted them in the manuscript.

Point 16: Typos and punctuations: The authors should have made a double check before submission.

Response 16: Thank you for your valuable advice! We have corrected the spelling mistakes and punctuation in our manuscript, and have checked the whole text repeatedly.

Point 17: Line 31: Synthetic is missed.

Response 17: Thank you for your valuable advice! We have revised the corresponding missing text and highlighted it in the manuscript.

Point 18: Line 357 is bold.

Response 18: Thank you for your valuable advice! We have revised the corresponding words in the manuscript and highlighted them in the manuscript.

Point 19: Lines 432 - 433) formulas are displaced.

Response 19: Thank you for your valuable advice! We have modified the corresponding formula in the manuscript and highlighted it in the manuscript.

Point 20: Lines 628-629) First, this & backbone, CRbackbone.

Response 20: Thank you for your valuable advice! We have corrected the corresponding grammatical errors in the manuscript and highlighted them in the manuscript.

Reviewer 2 Report

Please, see the attached pdf file.

Author Response

Response to Reviewer 2 Comments

Dear professor,

Thank you for reading the paper and for your careful comments!

Furthermore, We have completed relevant revisions and improvements to our paper based on your valuable comments.

Best regards,

Corresponding Author: Jie Chen1, 2, 3

1Information Materials and Intelligent Sensing Laboratory of Anhui Province, Anhui University Hefei, 230601, China

2 Key Laboratory of Intelligent Computing & Signal Processing, Ministry of Education, Anhui University, Hefei, 230601, China

338th Research Institute of China Electronics Technology Group Corporation in Hefei, China

Point 1: The description of the algorithm comes across as a little qualitative. I believe giving more quantitative detail would improve the paper.

Response 1: Thank you for your careful review and valuable comments! We have made a more qualitative addition to the description of the algorithm in the Algorithms section of the manuscript and modified the language of the algorithm, highlighted in Section 3.

Point 2: For the analysis of the state-of-the-art methodologies, the authors are invited to analyze also other interesting approaches [1]-[3] in introduction section.

[1] M.-S. Kang, et al, “Ground moving target imaging based on compressive sensing framework with singlechannel SAR,” IEEE Sensors J., vol. 20, no. 3, pp. 1238–1250, Feb. 2020.

[2] Lee, Myung-Jun, et al. "Improved moving target detector using sequential combination of DPCA and ATI."

The Journal of Engineering 2019.21 (2019): 7834-7837.

[3] Kim, Kyung-Tae, et al. SAR Image Change Detection via Multiple-Window Processing with Structural  

Similarity. Sensors. 2021, 21, 6645.

Response 2: Thank you for your careful review and valuable comments! We have presented three interesting state-of-the-art methods in the Introduction section and highlighted them in the manuscript, and cited the above references. The supplementary content is as follows:

For SAR target imaging problem,phase modulation from a moving target’s higher- order movements severely degrades the focusing quality of SAR images, because the conventional SAR Ground moving target imaging(GMTIm) algorithm assumes a constant target velocity in high-resolution GMTIm with single channel SAR. To solve this problem, a novel SAR-GMTIm algorithm[1] in the compressive sensing (CS) framework is proposed to obtain high-resolution SAR images with highly focused responses and accurate relocation. To improve moving target detector, research has propose a new moving target indicator(MTI) scheme[2] by combining displaced phase centre antenna(DPCA) and along-track interferometry(ATI) in sequentially to reduce false alarms compared to MTI via either DPCA or ATI. As shown from simulation results, the proposed method can not only reduce the false alarm rate significantly, but can also maintain a high detection rate and research has propose a synthetic aperture radar (SAR) change detection approach[3] is proposed based on a structural similarity index measure (SSIM) and multiple-window processing (MWP). [1] is the work proposed by focusing on SAR imaging. [2] The main work is to detect moving SAR targets. [3] The main work is to detect changes in SAR images, and our work is to focus on SAR target detection.

[1]. Kang, M.-S.; Kim, K.-T. Ground moving target imaging based on compressive sensing framework with single-channel SAR. IEEE Sensors Journal. 2019, 20, 1238-1250.

[2]. Lee, M.-J.; Kang, M.-S.; Ryu, B.-H.; Lee, S.-J.; Lim, B.-G.; Oh, T.-B.; Kim, K.-T. Improved moving target detector using sequential combination of DPCA and ATI. The Journal of Engineering. 2019, 2019, 7834-7837.

[3]. Kang, M.; Baek, J. SAR Image Change Detection via Multiple-Window Processing with Structural Similarity. Sensors. 2021, 21, 6645.

Point 3: The reviewer wonders if the authors’ method can provide the reliable performance for SAR target detection task in low SNR. Thus, it is suggested to test your proposed method in SNR simulation. In fact, it is one key issue for the robustness of the proposed method.

Response 3: Thank you for your careful review and valuable comments! We have tested whether our proposed method can provide reliable performance for SAR target detection SNR simulation in the experimental part, which is highlighted in the experimental part. The supplementary content is as follows:

To demonstrate the robustness of our proposed algorithm, We conducted comparative experiments with low SNR. We have conducted comparative experiments on salt and pepper noise, random noise, and Gaussian noise. In salt and pepper noise experiments, our method has mAP of 94.8, The map of our method is 5.5% higher than Yolo v3and 9.7% higher than Faster R-CNN. In random noise experiments, our method has mAP of 96.7, The map of our method is 3% higher than Yolo v3 and 11.1% higher than Faster R-CNN. In Gaussian noise experiments, our method has mAP of 95.8, The map of our method is 1% higher than Yolo v3 and 10.7% higher than Faster R-CNN. The experimental results show that we provide the reliable performance for SAR target detection task in low SNR. As shown in Tables 8. 

Point 4: The authors said that the discussed method can effectively address the SAR target detection task. What are the disadvantages of your method, computational cost? You did not mention anything in the test or experiment. This is unconvincing. Please explain it and add experiments to support your viewpoint. This is crucial to whether your paper can be accepted.

Response 4: Thank you for your careful review and valuable comments! We have explained and experimentally calculated the cost of our proposed method in the experimental section, which we list and highlight in the form of a table. The interpretation and experimental results are as follows:

In order that our proposed method can effectively solve the SAR target detection task, we also made corresponding experimental comparisons for the computational cost of Swin Transformer. The FPS and parameter statistics of several representative target detection algorithms are shown in Table 9. Because the current Swin Transformer The overall architecture of Transformer is still relatively large, the large volume of Transformer is a general problem in this field,, and we plan to make further improvements in model lightweighting and compression at present and in the future, which is also the focus of our work. Compared with the single-stage target detection algorithm, our parameter is 34M higher than YOLO V3, the FPS is 28.5 lower, but the mAP is 1.9% higher than YOLO v3.

Compared with two-stage target detection, the parameter amount is 52M higher than Faster R-CNN, and the FPS is 11.5 lower. Compared with Cascade R-CNN, the parameter amount is 8M higher and the FPS is 4.5 lower. Compared with Cascade Mask R-CNN, the number of parameters is 19M higher and the FPS is 7.5 lower. But mAP is 8.5% higher than Faster R-CNN, 7.7% higher than Cascade R-CNN, and 6.8% higher than Cascade Mask R-CNN.

Point 5: From the experiments of this paper, it is seen that the size of the surveillance scene is very small. But in practical SAR imaging, we usually face the SAR target detection problem of large scenes. Please add some experiment to demonstrate SAR target detection performance of the proposed method in the large scene.

Response 5: Thank you for your careful review and valuable comments! We have added visualization results of SAR object detection performance in large scenes in Section 4.5. The supplementary content is as follows:

    To demonstrate the object detection performance of our proposed method in large scene. We select our self-built SMCDD dataset as the inference dataset. Our original data was obtained from 38th Research Institute of China Electronics Technology Group Corporation. Because it belongs to the military secret institution, we signed a confidentiality agreement with them. The original image of the large scene obtained by HISEA-1. But to further demonstrate the effectiveness of our method, we use sliced data of some large scenes with a size of 2048*2048.As shown in Figure 15, from the visualization results, it can be seen that figures (1) and (6) have missed detection three places, figures (2), (4) have one false detection.

Point 6: I think there are still some rooms for improving the language of this manuscript. Many grammatical and typographical problems should be revised. Please ask a native speaker for corrections in these cases.

Response 6: Thank you for your careful review and valuable comments! We have checked the manuscript several times and have revised and invited a number of native English-speaking experts to polish our paper. Modifications are highlighted in the manuscript.

Round 2

Reviewer 1 Report

Regarding the revisions made by the authors, I would suggest acceptance of the paper in the current form.

Reviewer 2 Report

Please see the attached PDF file.
